# Zebrafish Paralogs *brd2a* and *brd2b* Are Needed for Proper Circulatory, Excretory and Central Nervous System Formation and Act as Genetic Antagonists during Development

**DOI:** 10.3390/jdb9040046

**Published:** 2021-10-31

**Authors:** Gregory L. Branigan, Kelly S. Olsen, Isabella Burda, Matthew W. Haemmerle, Jason Ho, Alexandra Venuto, Nicholas D. D’Antonio, Ian E. Briggs, Angela J. DiBenedetto

**Affiliations:** 1Medical Scientist Training Program, Center for Innovation in Brain Science, Department of Pharmacology, University of Arizona College of Medicine-Tucson, 1501 N Campbell Ave., Tucson, AZ 85724, USA; Branigan@email.arizona.edu; 2Biological and Biomedical Sciences Program, Department of Microbiology and Immunology, University of North Carolina School of Medicine-Chapel Hill, 321 S Columbia St., Chapel Hill, NC 27516, USA; kelly_olsen@med.unc.edu; 3Department of Molecular Biology and Genetics, Weill Institute for Cell & Molecular Biology, Cornell University, 239 Weill Hall, Ithaca, NY 14853, USA; irb25@cornell.edu; 4Institute for Diabetes, Obesity, and Metabolism, Smilow Center for Translational Research, Perelman School of Medicine, University of Pennsylvania, Room 12-124, 3400 Civic Center Boulevard, Philadelphia, PA 19104, USA; mhaem@pennmedicine.upenn.edu; 5Robert Wood Johnson Medical School, Rutgers University, Clinical Academic Building (CAB), 125 Paterson St., New Brunswick, NJ 08901, USA; jh1431@rwjms.rutgers.edu; 6Department of Biology, East Carolina University, Greenville, NC 27858, USA; venutoa18@students.ecu.edu; 7Sidney Kimmel Medical College, Thomas Jefferson University Hospital, 1025 Walnut St. #100, Philadelphia, PA 19107, USA; ndd007@students.jefferson.edu; 8Department of Biology, Villanova University, 800 Lancaster Ave., Villanova, PA 19085, USA; ibriggs1@villanova.edu

**Keywords:** gene duplication, functional antagonism, gene interaction, pronephros, peripheral blood island, spinal interneuron, BET proteins, bromodomain, cell death, lineage specification

## Abstract

Brd2 belongs to the BET family of epigenetic transcriptional co-regulators that act as adaptor-scaffolds for the assembly of chromatin-modifying complexes and other factors at target gene promoters. Brd2 is a protooncogene and candidate gene for juvenile myoclonic epilepsy in humans, a homeobox gene regulator in Drosophila, and a maternal-zygotic factor and cell death modulator that is necessary for normal development of the vertebrate central nervous system (CNS). As two copies of Brd2 exist in zebrafish, we use antisense morpholino knockdown to probe the role of paralog Brd2b, as a comparative study to Brd2a, the ortholog of human Brd2. A deficiency in either paralog results in excess cell death and dysmorphology of the CNS, whereas only Brd2b deficiency leads to loss of circulation and occlusion of the pronephric duct. Co-knockdown of both paralogs suppresses single morphant defects, while co-injection of morpholinos with paralogous RNA enhances them, suggesting novel genetic interaction with functional antagonism. Brd2 diversification includes paralog-specific RNA variants, a distinct localization of maternal factors, and shared and unique spatiotemporal expression, providing unique insight into the evolution and potential functions of this gene.

## 1. Introduction

Brd2 is a member of the bromodomain-extra terminal domain (BET) family of proteins, which act as epigenetic transcriptional co-regulators and are major interpreters of the acetyl-lysine-histone code [1,2]. BET proteins recognize and bind acetylated histones via their N-terminal bromodomains and recruit other proteins via their C-terminal ET domains [3]. Thus, they function as adaptor-scaffolds at target gene promoters, providing a surface for the ordered assembly and regulation of various chromatin-modifying factors and bridging transcription factors at distal elements with the basal transcriptional machinery [3,4]. Ultimately, BET proteins act to maintain activated or repressed states of gene expression via signal-dependent modulation of the chromatin structure [5]. Not surprisingly, BET proteins have been mapped at enhancer sites across the genome [6], are found at important nodes of interaction within the proteome [7], and are implicated in multiple biological processes including: meiosis and germline formation [8,9], *homeobox* (*Hox*) gene regulation [10], lineage specification [11,12], hematopoiesis [13], neurogenesis [14], cell cycle progression [15], maintenance of pluripotency [16], and, significantly, disease states such as cancer [17,18], kidney disease [19], metabolic syndrome [20], and neuroinflammation [21]. Consequently, small molecule inhibitors of BET proteins (BETi) have not only revealed BET-regulated transcriptome and proteome changes underlying different developmental processes, but also, are potential therapeutic agents for multiple diseases [22]. As master regulators of multiple target genes, BET proteins likely facilitate crosstalk among different regulatory pathways and biological processes, and act as critical relay stations for the integration of multiple signals [6,7]. A full understanding of how these kinds of regulatory networks work, how they grow more complex, and how they impact evolutionary and developmental trajectories, awaits parallel comparative study of gene function and interaction in multiple species. The study of BET protein Brd2 across species in the context of embryonic development illustrates this point.

The role of *Brd2* in development has been studied in Drosophila, mouse, and zebrafish, providing a rich source for functional comparison. In each of these species, *Brd2* produces both maternal and zygotic gene products necessary for survival and normal embryogenesis, suggesting an ancient, conserved role in germline formation, egg-to-embryo transition, and/or early embryo developmental events [23,24,25]. In addition, induced deficiencies of Brd2 during development demonstrate its necessity for the proper patterning, differentiation and morphogenesis of body segments in invertebrates [4,23,26], and of segmented tissues, most prominently the central nervous system (CNS), in vertebrates [27]. In Drosophila, ortholog Fsh1 acts as an upstream regulator of various segment-determining *homeobox *genes and functions in concert with Trithorax group (Trx-*G*) proteins, which form the major activating epigenetic axis in metazoan development [4,23,28]. For instance, Fsh1 interacts with distal-bound Trx-G chromatin activators at the proximal promoter of *Hox* gene *Ultrabithorax* (*Ubx*), aiding interaction with the basal transcriptional machinery and thereby maintaining proper *>Hox* expression domains in the trunk [4]. A Brd2/Hox regulatory pathway may in fact be evolutionarily conserved, as a deficiency in the zebrafish ortholog Brd2a results in misexpression in the developing brain of *homeobox* gene *eng2a* and *paired box* gene *pax2a*, both of which are known targets of Fsh1 in Drosophila [27]. Brd2a deficiency in zebrafish embryos also results in the mispairing of hindbrain (HB) and ventral nerve cord (VNC)-derived spinal interneurons on either side of the midline, suggesting that Hox-dependent patterning of the CNS in this region is defective [27]. In addition, Brd2 is necessary for Hox-dependent patterning and differentiation in hematopoietic tissue in zebrafish and mice [29,30]; whether Brd2 affects the formation of other non-neuronal Hox-dependent tissues needs further investigation.

In contrast to these conserved roles, the regulation of neuronal apoptosis appears to be a recently derived and specifically vertebrate developmental function of Brd2, as deficiency results in dramatic increases in cell death in the embryonic CNS of mice and zebrafish, but not flies [27,31,32]. *Brd2* knockout mice display excess apoptosis, impaired growth, and brain and neural tube defects [31], and, in other studies, exencephaly and significant changes in the expression of neurogenesis genes [32]. In zebrafish embryos, sublethal knockdown of Brd2a results in excess apoptosis during segmentation, when the CNS is undergoing differentiation and morphogenesis, leading to reduced hindbrain, undefined midbrain-hindbrain boundary region, and deformed spinal cord [27]. Surprisingly, the deficiency of Brd2 has little overall effect on mitosis in mouse or fish embryos [27,31,32], despite the fact that it is a known protooncogene and facilitator of the E2F-dependent transactivation of cell cycle genes in adult mammalian tissues [33]. Indeed, mammalian *Brd2* was known early on to be upregulated in some forms of human B cell lymphomas and leukemias; subsequently, its forced overexpression in the lymphoid compartment in transgenic mice was shown to lead to these same blood cancers [34]. The pro-mitotic function of Brd2 might thus be prominent in adult tissues, but play a more restricted specialized role in embryogenesis. Since Brd2 shuttles into the nucleus of mouse neuronal precursors during both mitotic and apoptotic events in normal development [35] and is necessary for the cell cycle exit and differentiation of mouse neuroepithelial cells in vivo [36], it likely plays a role in cell fate decisions between division, differentiation, and death, at least in neuronal populations. In any case, Brd2 appears to be a dual regulator of both apoptosis and mitosis in vertebrates, depending on the context. Brd2 is also implicated in human disease states that are relevant to its role in neuronal development, including neurodegeneration following stroke [37], and the defective neurogenesis underlying juvenile myoclonic epilepsy (JME) [38].

Because the zebrafish genome contains two copies of the *Brd2* sequence [25], comparative studies of the resultant paralogs can provide a unique window into the developmental functions, interactions, and recent evolution of this important gene. As summarized above, we previously described the developmental role of the *Brd2* ortholog in zebrafish, *brd2a* [27]. We also showed that both zebrafish paralogs, *brd2a* and *brd2b*, encode maternal and zygotic products, and are expressed in largely (but not exclusively) overlapping domains in the embryo, becoming enriched in the developing central nervous system and ventral trunk during segmentation stages [25]. On the other hand, they differ strikingly in the localization of maternal RNAs in the oocyte, in the RNA variants expressed in the embryo, and in genomic sequence and structure [25]. In particular, paralog *brd2b* produces a short RNA splice variant that potentially encodes a truncated protein carrying only bromodomain 1 (BD1). This suggests a dominant negative function for this isoform, leading us to wonder whether the two paralogs might be functional antagonists [25]. The work of others identifying *brd2b* as a tumor suppressor gene in zebrafish—opposite to the known protooncogene function of the human ortholog of *brd2a* [39]—supports this idea. In this study, we examine the developmental function of zebrafish paralog *brd2b* using antisense morpholino knockdown, and compare the findings to what we already know about the developmental role of *brd2a*, the ortholog of human *Brd2*. We uncover novel functions for Brd2b in the formation of the circulatory and excretory systems, and most significantly, demonstrate the occurrence of genetic interaction with functional antagonism between the two zebrafish paralogs during development. Our study is likely to inform a deeper understanding of Brd2 activity in mammals, and it also provides a concrete example of recent gene duplication and divergence in vertebrates, illustrating possible mechanisms of paralog diversification and interaction that may underlie the complexification of gene regulatory networks during development over evolutionary time.

## 2. Materials and Methods

### 2.1. Fish Maintenance and Handling

Adult wildtype zebrafish (Danio rerio, strain AB, University of Oregon, Eugene OR) were maintained at 28.5 °C on a 14 h light:10 h dark cycle and fed twice daily with a mixture of live brine shrimp and Zeigler Adult Zebrafish food (Pentair-AES, Cary, NC, USA). Embryos were generated from natural crosses in mating boxes, kept in E3 medium at 28.5 °C before and after microinjection, and staged by standard morphological criteria [40]. All experiments were conducted in accordance with protocols approved by the University Committee on the Use and Care of Animals (IACUC), Villanova University.

### 2.2. RNA Isolation and RT-PCR

For RNA isolation, ovaries were dissected and treated with Follicle Dissociation buffer (FDB; 400 µg/mL collagenase in Hank’s) for 10 min. After washing in Hank’s, ovaries were placed in ice-cold PBS for oocyte separation by stage. Embryos were dechorionated in E3 medium, incubated at 28.5 °C, and collected at selected stages using characteristic features according to Kimmel et al. [40]. RNA from oocytes and embryos was isolated using TRIzol reagent (Thermo Fisher Scientific, Waltham, MA, USA) according to company protocol. Briefly, 25 oocytes or 50 embryos per stage were homogenized in 1 mL of TRIzol in a 1.5 mL microfuge tube using a motorized pestle, with subsequent addition of 200 µL chloroform. Samples were gently mixed, incubated at room temperature for 2 min, and centrifuged at 12,000× *g* for 15 min at 4 °C. The aqueous layer was transferred to a fresh tube with 500 µL of isopropanol, incubated at room temperature for 10 min, and then centrifuged as before. The pellets were washed in 1 mL of 75% ethanol, centrifuged at 7500× *g* for 5 min at 4 °C, air dried for 10 min, and resuspended in 100 µL of RNase-free water. RNA samples were then purified using the Qiagen (Germantown, MD, USA) RNeasy kit with on-column DNase treatment, according to company protocol. The final RNA samples were quantified using a Nanodrop spectrophotometer prior to cDNA synthesis.

The cDNA template for RTqPCR was synthesized using the iScript cDNA synthesis kit (Bio-Rad), according to company protocol. Briefly, for each sample, 1.5 µL of RNA (typically 0.75 µg) was added to 13.5 µL iScript RNase-free water and 4 µL iScript reaction mix, heated to 65 °C for 2 min, then supplemented by 1 µL iScript reverse transcriptase and incubated in a Bio-Rad thermal cycler with heated lid as follows: 42°C for 60 min, 85 °C for 5 min, and 4 °C hold. A quantity of 20 µL of nuclease-free water was added to each sample and aliquots were stored at −20 °C until use.

To assay for *brd2b-L* and *brd2b-S* mRNAs in vivo, exon–exon junction-spanning primers were designed for exon pairs 5/6 (forward 5′-CCAGATTCGCAGTTCTCCAC; reverse 5′-CTCTTCGGGAGGGCTGAT), 9/10 (forward 5-GGGACAAACCTGCCAAACT; reverse 5′-GCAGTGAAGGCTCTCTGGAC) and 11/12 (forward 5′-GTCGAGAGGAGTTGGCTCTG; reverse 5′-GATTCCACGGTTTCCCATTA) of the predicted full-length transcript (Ensembl_Trans:ENSDART00000148709, brd2b-202), and the exon5/intron5 junction (forward 5′-CACGCGCACACACTTTTTAT reverse 5′-TTTAGTTTGCAGGGCGAGAG) of the shorter truncated transcript (NM_001110524.1). RTqPCR was performed on cDNA from the RNA of staged oocytes (stages I–IV) and embryos (4, 24, 48 hpf). Primers for positive control β-actin (forward 5′-GCAGAAGGAGATCACATCCCTGGC; reverse 5′-CATTGCCGTCACCTTCACCGTTC) and for negative genomic control intron 1 (forward 5′-CAGTGTCTGTTTTTGGCTCGG; reverse 5′-GGTCTCATTTTCCCCCAAAAGC) were also used in each experiment. RTqPCR was performed using Qiagen SYBER Green Real-Time PCR master mix. For each sample and each primer pair, 3 µL of five-fold diluted aliquots of cDNA were mixed with 10 µL of Evergreen master mix, 0.2 µL of 100 µM forward primer, 0.2 µL of 100 µM reverse primer, and 6.6 µL of nuclease-free water, in a well of a 96-well dish. Samples were placed in a real-time PCR detection system (CFX Connect Bio-Rad, Hercules, CA, USA) for 39 cycles of the following protocol: 95 °C, 5 min; 95 °C, 30 s; 58 °C, 30 s; and 78 °C, 3 min. The expected size and sequence of PCR products was confirmed by gel electrophoresis and cycle sequencing, respectively.

### 2.3. Brd2b Peptide Antibody Production, Western Blot and Peptide Competition

Anti-Brd2b peptide polyclonal antibody was generated in rabbits by Genescript (Piscataway, NJ, USA) against the antigenic determinant KSSRASLSSSQSKK, which was assessed by Blastp to be unique to Brd2b in the zebrafish proteome and is not present in the paralog (Brd2a) or other BET family members. Protein bands detected by the anti-Brd2b peptide antibody are erased by peptide competition, indicating specificity of the antibody for the antigenic determinant, and presence of the peptide sequence in all detected proteins (see below). As the peptide epitope resides just before the ET domain, only the full-length isoform Brd2b-L, and not Brd2b-S, can be detected by this peptide antibody.

For western blot analysis, oocytes and embryos were collected as described above for RTqPCR, and samples were homogenized in 1.5 mL microfuge tubes in 45 µL of RIPA buffer (50 mM Tris-Cl pH 8, 150 mM NaCl, 0.5% sodium deoxycholate, 1% NP40, 0.1% SDS) with added Roche mini protease inhibitor cocktail (Millipore-Sigma, Burlington, MA, USA) and rotated at 4 °C for two hours. After centrifugation at 12,000 rpm for 15 min to pellet debris, supernatants were transferred to fresh tubes, 45 µL of 2× SDS loading buffer was added, and tubes were vortexed and placed at 100 °C for 5 min, before sample loading on 8% polyacrylamide gels (Thermo Fisher Scientific, Waltham, MA, USA) and electrophoresis at 100 V for 50 min. Transfer to nitrocellulose membranes was conducted at 40 V for 90 min at room temperature in a Bio-Rad mini-Protean transfer unit with an ice pack. To assess equal loading across lanes, all membranes were stained after transferring with the Memcode Reversible Protein Stain (Thermo Pierce, Waltham, MA, USA) and imaged to visualize total protein before antibody incubations, as per company protocol. This process was undertaken in lieu of reprobing blots with antibodies to a control “housekeeping” protein, as this was found to be problematic in terms of efficiency of binding in reprobed membranes. Membranes were rinsed in PBST, blocked in PBST/5% dry milk at room temperature for 2 h, then placed in 6 mL PBST/1% dry milk containing 12 µL anti-Brd2b primary antibody (1/500 dilution) and incubated at 4°C on shaker overnight. After washing in PBST, membranes were incubated in 10 mL PBST/1% dry milk with 3 µL of goat anti-rabbit-HRP secondary antibody (Pierce) at room temperature for 2 h, with shaking. After PBST washes, membranes were incubated in Supersignal West Dura detection buffer (Thermo-Fisher Scientific, Waltham, MA, USA) for 5 min and imaged by means of chemiluminescence. Precision Plus Protein dual color and WesternC molecular weight markers (Bio-Rad, Hercules, CA, USA) were used as size standards.

For peptide competition, duplicate blots of identical samples were prepared, and prestaining with Memcode was conducted as above. One blot was incubated with anti-Brd2b primary antibody; the second blot was incubated with anti-Brd2b primary antibody that had been first pre-incubated with 15 µM of reconstituted peptide (Genescript, Piscataway, NJ, USA) in PBS for 2 h at room temperature. Secondary antibody treatment and detection was conducted as above for both membranes. Brd2b protein expression differences between the MO-treated and control embryos were estimated by measuring band pixel densities for each sample lane on blots using ImageJ FIJI, and then standardizing to total protein loaded for that lane. Total protein loaded per sample was estimated by measuring and summing pixel densities for the visible Memcode bands per lane/sample.

### 2.4. Full-Length cDNA Cloning

A cDNA clone of *brd2a* was previously obtained in a screen of a cDNA library from 15 to 18 hpf zebrafish embryos and was subcloned directly into the pCS2+ vector. cDNA sequences for *brd2b-L* and *brd2b-S *were obtained by RTqPCR from embryonic RNA using primers from the *brd2b* predicted sequences in the ZFIN database (Ensembl_Trans:ENSDART00000148709, brd2b-202). Primers for *brd2b-L*: forward 5′-AGGTGCAGAATAGCAATAAAACG; reverse 5′-GCAGTTCCTAAACTCTCATTACCTTC. Primers for *brd2b-S*: forward 5′-GGTGAGCTCGAGCATAAGGT; reverse 5′-TGCCAATTTAGTTTCAATAGTTCA. PCR products were first cloned into pCR 2.1 TOPO vector and inserts representing both *brd2b * short and long cDNAs were confirmed by sequencing. Both long and short *brd2b* cDNAs were then subcloned into pCS2+ vector by PCR using primers designed to exclude the *brd2b*MO2 binding site. The pCS2+ vector contains transcriptional start and stop sequences, and a poly-adenylation signal, and allows for in vitro transcription of sense RNA for use in rescue-enhancement studies described below.

### 2.5. Antisense Morpholino Treatments

Gene-specific antisense morpholino oligonucleotides (MOs) were purchased from GeneTools, LLC (Philomath, OR, USA). Stock MO solutions (1 mM) in sterile glass distilled water were diluted to various working concentrations in 1× Danieau buffer and 0.15% phenol red, and a constant volume of 2 nL MO working solution was injected into each 1–4 cell stage embryo using a Leitz micromanipulator (Leica Microsystems, Inc., Wetzlar, DEU) and a PV820 pneumatic picopump (World Precision Instruments, Sarasota, FL, USA)). Two independent *brd2b*-specific MOs targeting non-overlapping regions of the 5′UTR and/or region surrounding the initiation codon of zebrafish *brd2b* were designed with company help: 2bMO1 (5′-CTTTGCGGTGCGAAATCCTGCGTTT-3′), targeting 25 bases of 5′UTR 18 bases upstream of the putative start codon; and 2bMO2 (5′-ATTGCTATTCTGCACCTTCTGCTCG-3′), targeting 25 bases of 5′UTR upstream of MO1. The uniqueness of each MO target was verified by Blastn searches of the zebrafish transcript and genomic databases at NCBI. A standard negative control MOc supplied by the company (5′-GGTTTCGCTCGAATATCCGAGTTTC-3′), and a five base-pair mismatch to 2bMO1 (2bMO1mis5; 5′-CTTTcCGcTGCcAAATaCTGCcTTT-3′) were used to test for general non-specific toxic effects and effective range dosage, respectively. Morpholinos against *brd2a* were previously described and authenticated [27].

To determine the effective range of morpholino dosage, working 2bMO solutions between 0.5 and 8 ng/nL were tested on a total of 25–40 embryos per treatment for brain defects at prim 5. An MO concentration was deemed to be in the effective range if it caused characteristic and easily identifiable defects in *brd2b* morphants compared to wildtype, while the standard control morpholino (MOc) and five-base mismatch (2bMO1mis5) at the same concentration caused no effect. Using these criteria, injected *brd2b*MO1 amounts from 1 to 6 ng were judged to be within the effective range; *brd2b*MO2 was then tested in this same range. The incidence of brain defects varied significantly both by treatment within each dosage in the effective range (*p* < 0.0001, chi square contingency and Fisher’s exact test), and by dosage (*p* < 0.0001, Cochran–Mantel–Haenszel test). Differences between uninjected “none” and “MOmis5” treatments were not statistically significant, either within each dosage or across dosages, while differences between “2bMO1” and either “none” or “MOmis5” were significant (Correspondence analysis). For experiments in this paper, 2 ng 2bMO total amounts were used, unless indicated otherwise.

For initial RNA rescue experiments, sense mRNA lacking *brd2b*MO binding sites was synthesized from linearized pcDNA1-Amp vector carrying HA-tagged human Brd2 (NM_005104.4) using the mMESSAGE mMACHINE T7 Capped RNA transcription kit (Ambion Inc., Austin, TX, USA). Synthesized mRNA was sized and quantified by standard formaldehyde gel electrophoresis. Embryos were co-injected with 2 ng of either *brd2b*MO1 or *brd2b*MO2, and different amounts of sense RNA ranging from 50 to 500 pg, before selecting 250 pg as the standard for co-injections. For co-injections with antisense p53 morpholinos (*p53*MO, GeneTools, Philomath, OR, USA), each morpholino alone (2–4 ng *brd2b*MO1; 4 ng *p53*MO), and the combination (4:4 ng *brd2b*MO1:*p53*MO; or 2:4 ng *brd2b*MO1:p53MO) was injected. For co-knockdown of Brd2b and Brd2a, a mix of *brd2a*MO (4 ng) and *brd2b*MO (2 ng) morpholinos was injected. We detected no sequence complementarity between any of the *brd2a* and *brd2b* morpholinos we used, and we obtained the same suppressor effects; however, we mixed MO1 or MO2 morpholinos from each locus. In addition, we conducted sequential (rather than mixed) injections in subsequent experiments and obtained the same co-knockdown effects.

For rescue-enhancement studies, sense RNA was synthesized from cloned zebrafish *brd2a*, *brd2b-L* and *brd2b-S* cDNAs (see full-length cDNA cloning above) using the mMESSAGE mMACHINE Sp6 Capped RNA transcription kit (Ambion Inc., Austin, TX, USA). Working solutions were made by diluting RNA to 100ng/µL using in 0.1 M KCl, embryo-tested water (Millipore-Sigma), and 0.1% phenol red. A quantity of 2 nL of working solution was microinjected into embryos at the 1–4 cell stage for all rescue and enhancement studies, since at this dosage all three synthetic RNAs injected on their own produced a mild phenocopy of the morphant phenotype but did not result in uncharacteristic or severe defects. For rescue studies, morpholinos were injected in 4 ng amounts, while for enhancement studies, morpholinos were injected in amounts of between 2 and 4 ng to allow for enhancement to be detected more easily.

### 2.6. Crispr-Cas9 Disruption

Clustered regularly interspaced short palindromic repeats (CRISPR) crRNA was designed against the *brd2b* gene on chromosome 16 using CRISPRscan (http://www.crisprscan.org, accessed on 16 June 2016 and purchased along with tracerRNA (trRNA) and Cas9-mRNA from GE Health Dharmacon Inc. (Lafayette, CO, USA) The following *brd2b* RNA coding sequence was targeted from NM_001110524.1, Exon 2: GUGGCUCGUCCUUGUCGGCUGUUUUAGAGCUAUGCUGUUUUG. Stock solutions of crRNA (1 ng/μL in TrisCl pH 7.4 RNase free) were added at equal concentrations to trRNA and incubated on ice with 2X concentration of Cas9-mRNA for 5 min. The CRISPR solution (crRNA + trRNA + Cas9-mRNA) was then injected into the animal pole of 1 cell stage embryos using a Leitz micromanipulator (Leica Microsystems, Inc. Wetzlar, DEU) and a PV280 pneumatic picopump (World Precision Instruments). The efficiency and specificity of *brd2b* crRNA was assessed by PCR cloning followed by mismatch detection using T7 endonuclease I assay (Guide-It Mutation Detection kit, Takara Bio/ClonTech, Mountain View CA, USA). Briefly, five uninjected and five Crispr-injected embryos at prim 5 stage were incubated in 100 μL of 50 mM NaOH for 10 min at 95 °C, then cooled to 4 °C and neutralized with 1/10 volume of 1 M Tris-Cl, pH 8.0. Debris was removed was removed by centrifugation and 5 μL of supernatant was used in PCR, according to kit instructions. Primers for amplification of the targeted *brd2b* region were selected using Primer 3: forward 5′-CGAGTTCCGTTCATACCAATC and reverse 5′-TCGAAACTGTGTCAAT-CCAGA. PCR products were denatured and allowed to anneal slowly before digestion with the Resolvase enzyme from the kit and gel electrophoretic analysis. Crispr-Cas9 efficacy was estimated using ImageJ FIJI by comparing pixel density of intact PCR product bands from Crispr-disrupted thouembryos, with and wit resolvase treatment. Typically, a total of 20–30 embryos (F0) was used per treatment group in each experiment and were scored for morphological brain defects characteristic of *brd2b* morphants, as well as assayed by Terminal Transferase dUTP Nick End Labeling (TUNEL) for cell death levels.

### 2.7. In Situ Hybridization

*In situ* hybridizations to RNA in whole mount zebrafish embryos were conducted according to the Schulte–Merker protocol as found in the Zebrafish Book [41], with slight modification. Acetic anhydride and proteinase K treatment were used on older embryos only (>18 hpf). Digoxigenin (DIG) labeled RNA probes were prepared according to company instructions (Roche Biochemicals, Basel, CHE), and were used directly in hybridizations without prior hydrolysis. Hybridizations were conducted with 1 ng/mL probe in 50% formamide buffer with 5 mg/mL torula yeast, type VI (Sigma, St. Louis, MO, USA), and without heparin, at 65 °C overnight. Post-hybridization washes (50% formamide, 2× SSCT; 2× SSCT; 0.2× SSCT) were carried out at 55 °C, without RNase treatment. The embryos were blocked in PBST plus 10% FBS, 0.1% Tween-20, and 1% DMSO for 4 h before anti-DIG detection in staining buffer plus 1 mM levamisol. After detection, the embryos were fixed overnight in 4% paraformaldehyde, then dehydrated and stored in methanol at −20 °C. For imaging, embryos were cleared in 2 parts benzylbenzoate: 1 part benzylalcohol and mounted in Canada balsam. For In situ examination of ovaries, the high-resolution whole-mount protocol of Thisse, as described in The Zebrafish Book [41], was followed exactly. Images of 5–7 representative embryos were taken on a dissecting scope fitted with a Leica DC camera (Leica Microsystems, Wetzlar, DEU)).

### 2.8. Immunohistochemistry

For immunohistochemistry, staged embryos or dissected ovaries were fixed in 4% paraformaldehyde in PBS overnight at 4 °C, then washed in PBS, permeabilized in an ascending methanol/PBS series and stored in 100% methanol at −20 °C. Embryos that were 24 hpf or older were dechorionated before, and younger embryos after, fixation. For the Pax2a experiments, samples were rehydrated in a descending methanol/PBS series, treated with 24 µg/mL proteinase K/PBS for 5 min, washed again in PBS, incubated for one hour at room temperature in block solution (5% normal goat serum, 3% TritonX, 1× PBS), and then incubated in a 1: 200 dilution of rabbit anti-Pax2a primary antibodies (GeneTex, Irvine, CA, USA) in blocking solution overnight at 4 °C. After washing in PBS, samples were incubated in goat anti-rabbit Alexa555 secondary antibodies (Vector Laboratories, Burlingame, CA, USA) in PBS for one hour at room temperature, and washed again in PBS. Embryos were mounted on glass slides with Vectashield with DAPI stain (Vector Laboratories, Burlingame, CA, USA) and imaged by conventional fluorescence (Leica DM LMD), or laser-scanning confocal (Leica TCS SP8) microscopy, and optical sections and 3D maximum projection images were obtained. Pax2a(+) cells in the pronephros were counted by manual tagging of lif files using ImageJ (FIJI) software. Pax2a(+) interneurons were analyzed for pairing on either side of the central verve cord by counting the number of symmetrically opposed interneuron pairs and the number of interneurons appearing solo on either one or the other side of the nerve cord, and comparing these numbers in uninjected control and morpholino-injected embryos. Three-dimensional maximum projection confocal images of the dorsal spinal cord from 3 to 6 samples per treatment group were analyzed by ImageJ FIJI software and chi-square contingency tests.

### 2.9. TUNEL Analysis

Embryos were fixed overnight in 4% PF/1% DMSO/PBS at 4 °C, dehydrated through an ascending methanol/PBS series, and stored in methanol at −20 °C overnight. After rehydration, embryos were treated as per the ApopTag Fluorescein In Situ Apoptosis Detection Kit protocol (Millipore-Sigma, Burlington, MA, USA). Blocking was conducted for one hour in fetal bovine serum (FBS)/PBST, and embryos were incubated overnight with 100 μL working strength anti-digoxigenin-fluorescein conjugate at 4 °C in dark. Embryos were mounted in Vectashield on glass slides with footed coverslips for fluorescence imaging. Fluorescent TUNEL was visualized with a TCS SP8 laser-scanning confocal microscope (Leica Microsystems, Inc., Wetzlar, DEU), and 55–60 optical sections and 3D maximum projection images were quantitatively analyzed. Alternatively, a conventional DM LMD fluorescence microscope (Leica Microsystems, Inc., Wetzlar, DEU) was used, and 15 to 35 optical sections (depending on experiment) were analyzed. Cell counts were obtained by manual tagging of lif files using ImageJ (FIJI) software. Cell death counts from three to seven embryos per treatment, depending on experiment, were analyzed for statistical significance by one way ANOVA, ANOM (Analysis of Means) and Tukey’s HSD tests, using JMP Pro13 software.

### 2.10. Phenotypic Assessment and Population Data

Phenotypic assessments were carried out at a population level to assess expressivity and penetrance. Typically, a total of between 30 and 60 embryos were used per treatment group in each morpholino experiment (MO knockdown, RNA rescue, p53MO co-injection, suppression-enhancement studies) unless noted otherwise, and were scored for morphological brain and trunk defects. Brain defects were also rated mild, moderate, and severe in some experiments to assess expressivity. Phenotypic population data were analyzed for significant differences between treatments by chi square contingency, Fisher’s exact tests and Correspondence analysis. For TUNEL, In situ hybridization (ISH), and the immunohistochemical (IHC) assays, a total of 20–50 embryos per treatment, depending on experiment, were first examined qualitatively for characteristic morphological defects, then 3–10 representative embryos were selected for imaging and quantitative analysis by one way ANOVA, ANOM, and Tukey’s HSD tests, using JMP Pro13 software.

## 3. Results

### 3.1. Zebrafish brd2b Exhibits Transcript Variants Differentially Regulated during Development

We originally cloned zebrafish *brd2a* and *brd2b* coding sequences from a cDNA library made from 15 to 19 h post fertilization (hpf) embryos, using a degenerate probe derived from the conserved first bromodomain of mammalian Brd2 [25]. The zebrafish *brd2a* cDNA encoded a canonical full-length BET protein (3.9 kb, 836 aa; NP_001257500.1), orthologous to human Brd2, containing two N-terminal bromodomains (BD1, BD2), a nuclear localization signal (NLS), and a C-terminal extraterminal (ET) protein-interaction domain [1]. The *brd2b *cDNA, however, appeared to be a truncated sequence (1.4 kb) from a paralogous locus, derived from an alternatively spliced mRNA that retained intronic sequences at the exon5/intron5 splice junction and terminated within that intron (Figure 1B, *brd2b-S*; NM_001110524.1) [25]. The retained intron introduced a premature stop codon to the protein coding sequence. Thus, when conceptually translated, this short transcript variant (*brd2b-S*) would produce a truncated Brd2b protein (Brd2b-S) of 276 amino acids, having only a single bromodomain 1 (BD1), and lacking the second bromodomain (BD2), NLS, and ET domain characteristic of BET proteins (NP_001103994.1). As a full-length Brd2b protein (Brd2b-L in this paper) is predicted algorithmically (Figure 1A, Brd2b-L, 811 aa; Figure 1B, *brd2b-L*, 4037 nt; transcript—Ensembl_Trans:ENSDART00000148709, brd2b-202, Ensemble release GRCz11), the existence of a truncated version (Brd2b-S) suggested to us a possible dominant negative function, whereby it might compete with Brd2b-L and/or its paralog Brd2a for chromatin target sites, while being itself “cargo-less” due to the lack of the protein interaction ET domain. To verify that *brd2b-S* was an actual in vivo transcript rather than a cDNA library cloning artifact, and to empirically confirm the presence of predicted exons of the longer transcript (*brd2b-L*), as found in the present databases, we designed primer pairs spanning diagnostic exon–exon junctions in the two cDNAs, and conducted RT-PCR on RNA isolated from stage I–IV oocytes and from 4, 24, and 48 h post fertilization (hpf) embryos (Figure 1B, exon 5/6, exon 9/10, exon 11/12 primers from *brd2b-L*, and exon5/intron5 primers from *brd2b-S*; Figure 1C, key of primer pairs and their predicted products; gels showing RT-PCR products obtained). *brd2b-L* exons are present in all tested oocyte stages and embryos (Figure 1C, blue arrowheads, lanes labeled 3, 4, and 5, all sample gels). We also observed PCR product from exon5/intron5 primers diagnostic for the alternative *brd2b-S* transcript, which we verified by sequencing. Strikingly, in contrast to the full-length RNA, *brd2b-S* predicted products are detected unquestionably only in later stage embryos (Figure 1C, red asterisks, lane labeled 2, for 24 and 48 hpf sample gels only). The faint bands visible in lanes labeled 2 for stage 1,2,3 oocyte samples are either smaller than the predicted *brd2b-S* product (s1, s2 gels) or among multiple very faint bands in a lane (s3 gel), leading us to consider them spurious. Nevertheless, we cannot rule out completely the possibility that the very faint band near the correct size (s3 gel) might represent *brd2b-S* at very low levels, although we consider this unlikely. Importantly, primers derived exclusively from within intron 1 show no product (Figure 1C, lane labeled 9 for each sample gel), ruling out genomic DNA contamination as a source template. These data corroborate our previous analyses by Northern blot, where we detected *brd2b* RNA variants of corresponding sizes, with longer RNAs (4 and 6 kb) observed in ovaries and embryos of all stages, and a short mRNA of about 1.8 kb only in embryos starting between 8 and 10 hpf and persisting through 48 hpf [25]. Thus, both *brd2b-L* and *brd2b-S* transcript variants are detected in vivo, and show distinct tissue and temporal patterns of expression.

### 3.2. brd2b Encodes a Maternal/Zygotic Factor That Is Localized to the Animal Pole in Oocytes and Enriched in the CNS and Ventral Trunk in Embryos

To study the protein products of the *brd2b* locus, we generated antibodies against zebrafish Brd2b using synthetic peptides derived from the predicted amino acid sequence and verified by Blastp to be unique in the zebrafish genome. The first peptide epitope we targeted resides between the NLS and the ET domain, so derived antibodies would detect the predicted Brd2b-L protein isoform but not putative Brd2b-S. Using this antibody in western blot analyses, we found a major band of >150 kD consistently detected in ovaries/oocytes and in 48 hpf embryos, but inconsistently and at very low levels at 24 or 4 hpf (Figure 2A). Another major band between 50 and 75 kD is consistently detected in embryos at 48 hpf, less prominently at 24 hpf, sometimes faintly at 4 hpf, and rarely in ovaries/oocytes (Figure 2A; Appendix A). All bands are subtracted by peptide competition (Figure 2, compare A with C), indicating detected proteins contain the peptide epitope and are likely Brd2b isoforms or breakdown products. Prestaining of blots for total protein (Figure 2B,D) shows that lack of signal on blot in C is due to peptide competition during antibody incubation and not due to lack of protein loading. It is possible that the band at >150 kD represents full-length Brd2b-L (predicted MW 91 kD), where differences from the predicted might be due to post-translational modification and/or stable complex formation [42]. In any case, the putative full-length protein is found in both ovaries/oocytes and embryos, while the smaller product is most prominently found in later stage embryos. In addition, the presence of both the *brd2b-L* mRNA and the putative Brd2b-L protein in both ovaries/oocytes and embryos supports the correspondence of this pair as maternal (and zygotic) products. We attempted to detect the putative short isoform Brd2b-S, using antibodies against an N-terminal epitope upstream of BD1 that was common to Brd2b-L and Brd2b-S, but this antibody was ineffective and unable to detect either isoform in any sample.

Immunohistochemical analysis using the working anti-Brd2b peptide antibody shows the presence of maternal Brd2b protein in all stages of oocyte, with dramatic changes in localization over time (Figure 2E). Brd2b is dispersed throughout the cortical cytoplasm in stages I and II, gradually concentrated around the germinal vesicle (GV) by early stage III, and finally, by late stage III–stage IV when the GV breaks down, strikingly localized around the micropyle, a specialized follicle cell that marks the future animal pole. This protein localization pattern follows the previously noted track of cognate *brd2b* mRNA in staged oocytes, but differs dramatically from the path followed by paralogous *brd2a* mRNA and protein, both of which end up in the peripheral cortex just inside plasmalemma of the oocyte [25]. Thus, similarly to its paralog, Brd2b is spatially regulated during oogenesis, but it obtains a strikingly different endpoint position.

Finally, we used the anti-Brd2b peptide antibody to assess the efficacy of Brd2b antisense morpholino knockdowns (described in the next section) and saw reduced levels of Brd2b-L (>150 kD) by western blot (Figure 2F,G; ~3-fold reduction in morphants, ImageJ quantification; see the Figure 2 legend and the Materials and Methods Section).

### 3.3. Brd2b Knockdown Results in Reduced Hindbrain, Ill-Defined MHB, and Trunk Abnormalities Similar to Brd2a Morphants, but Presents Unique Circulatory and Pronephric Defects

We showed previously that a deficiency of Brd2a during zebrafish development results in malformation of the neural tube, somites, and post-anal trunk, reduced hindbrain and ill-defined midbrain-hindbrain boundary (MHB), mispairing of spinal interneurons, and abnormal numbers of Pax2a(+) cells in ventral trunk—all accompanied by a dramatic misregulation of cell death and of some of the genes necessary for proper formation of the MHB and anterior hindbrain [27]. Here, we conduct a comparative study to examine the effects of the deficiency of paralog Brd2b, using both antisense morpholinos and Crispr-Cas9 disruption. We designed two independent non-overlapping antisense translational block morpholinos (*brd2b*MO1 and *brd2b*MO2) to assess morphant phenotypes and used two control morpholinos (MOc and MOmis5) to control for general toxicity and determine effective dosage range for subsequent studies (see Materials and Methods).

We first surveyed visible phenotypes of Brd2b morphants at various stages of development after injection of 2–4 cell embryos with 2 ng of *brd2b*MO, typically examining a total of 40–60 embryos from three independent clutches per treatment. We injected 2 ng MO as it was the lowest dose in the effective range (1–6 ng) that consistently produced clear, mostly penetrant morphant phenotypes (see Materials and Methods). *brd2b*MO morphants begin to show defects in early segmentation stages, and by prim 5 characteristic head abnormalities are clearly seen, including reduced overall brain size, collapsed hindbrain ventricle, and reduced and/or ill-defined MHB region (Figure 3A vs. Figure 3B,D). Both *brd2b*MO1- and *brd2b*MO2-injected morphants give the same characteristic brain abnormalities (Figure 3B,D), as do embryos treated with Crispr-Cas9 to disrupt the *brd2b* gene (Figure 3E). Population data show statistically significant differences between control and experimental treatment groups for these studies (Table 1, *brd2b*MO single knockdown, *p* < 0.0001). Importantly, we obtained substantial rescue of *brd2b*MO morphant brain defects by co-injection with in vitro-synthesized human *Brd2* mRNA (*HsBrd2*RNA) lacking *brd2b*MO target sequences (Figure 3C; Table 1, *brd2b*MO1+/−*HsBrd2*RNA rescue, *p* < 0.0001). These *brd2b*MO morphant brain defects recapitulate what we found previously with knockdown of Brd2a (Figure 3F), except that they are typically more highly penetrant, and manifest reliably at lower MO doses (2 ng *brd2b*MO vs. 4 ng *brd2a*MO).

In addition to the morphant brain defects that were shared by the paralogs, we found novel abnormalities that were unique to Brd2b knockdown in the ventral trunk and circulatory system. By the prim 5 stage, *brd2b*MO morphants exhibit a malformed pronephric duct with the distal tip cloaca often clogged by a terminal plug of “extra” cells, rather than opening to the outside (Figure 3B,D,E vs. Figure 3A,C; trunk insets, top arrows; also, refer to diagram of pronephros in Figure 6N). In addition, the region just posterior to the duct and ventral to the nerve cord (see Figure 3A, pbi) is sometimes disorganized or clumpy compared to its uniform and granular appearance in wildtype (Figure 3B,D,E vs. Figure 3A,C; trunk insets, bottom arrows). This region is home to the peripheral blood island (PBI), where a transient wave of hematopoiesis is ongoing during segmentation [43]. Heartbeat in terms of rate and strength is impaired from prim 5 onward as well. These defects are phenocopied by Crispr-Cas9 disruption of *brd2b* (Figure 3E; trunk inset, arrows) and rescued substantially by co-injection with *HsBrd2*RNA (Figure 3C; trunk inset). Population data confirm that differences between treatment groups for trunk and heart defects are statistically significant (Table 1, *brd2b*MO single knockdown and *brd2b*MO1 +/− *HsBrd2*RNA rescue, *p* < 0.0001). Although *brd2a*MO morphants show post-anal trunk defects with disrupted often “vacant” PBI tissue at a higher frequency, they lack the clogged pronephric duct defect characteristic of *brd2b*MO morphants (Figure 3F vs. Figure 3B,D; trunk inset, upper arrows for pbi). As development proceeds, *brd2b*MO morphants display progressively slower heart rates, weaker heartbeats, and severe pericardial edema. Eventually, complete lack of circulation leads to degeneration of the trunk (Figure 3I,J) and death typically by 5–7 dpf. This contrasts with *brd2a*MO morphants, which can survive up to two weeks or more with continued circulation [27].

Similarly to what we observed in Brd2a knockdown studies [27], *brd2b*MO morphant defects show variable expressivity and, to a lesser degree, incomplete penetrance; both the severity and penetrance of defects increase with dose for *brd2b*MO1 (*p* < 0.0001, chi square contingency and Cochran–Mantel–Haenszel tests). For instance, at 0.5 ng *brd2b*MO1, 76.9% of morphants show wildtype brain, while 19.2% show mild and 3.8% moderate brain defects; at 2 ng *brd2b*MO1, 0% morphant brains are wild type, while 78.9% are mildly, 15.8% moderately, and 5.3% severely defective; at 6 ng, 0% morphants are wild type, 31.5% show mild, 52.6% moderate, and 15.7% severe brain defects. Pooling mild, moderate, and severe morphant phenotypes together, we typically see 100% penetrance of the brain defect, between 88 and 100% penetrance of the pronephric duct and circulation defects, and around 25% penetrance of the peripheral blood island defect, when using 2 ng of either *brd2b*MO (Table 1, *brd2b*MO single knockdown). Crispr-Cas9 disruption of the *brd2b* locus results in 100% penetrance and moderate to severe brain, duct, and circulatory defects, while PBI abnormalities remain at 21% penetrance and moderate to mild severity.

Taken together, these studies show that *brd2b *exhibits both shared and unique developmental functions with its previously studied paralog *brd2a,* in expression domains such as the germline and the embryonic brain and ventral trunk. Significantly, *brd2b* produces a tissue- and stage-specific transcript variant and possibly several protein isoforms, and is required uniquely for the proper formation of the pronephros and cloaca and for ongoing blood circulation in developing embryos.

### 3.4. Brd2b Knockdown Increases Cell Death in the CNS of Prim 5 Morphant Embryos but Reduces Cell Death in the Cloaca of the Pronephros

Since we knew that a deficiency of Brd2a leads to similarly reduced and defective brains as seen here, and also to dramatic increases in neural cell death [27], we examined levels of apoptosis in Brd2b-deficient morphants as well. A minimum of 30 embryos per treatment group were subjected to whole-mount fluorescent TUNEL assay at prim 5 followed by laser-scanning confocal microscopy to produce optical sections and maximum projection images. Six to ten representative embryos from each treatment group were then used in quantitative analysis of apoptotic nuclei. Three sets of tests were conducted to assess the authenticity of cell death in morphants: set 1, Rescue by HsBrd2 RNA, to test gene-specificity; set 2, Corroboration by independent means, to obtain the same effects without morpholinos; and set 3, p53-independence, to test for off-target apoptotic effects [44]. Representative maximum projection images from these experimental sets are shown in Figure 4 and Appendix A and accompanying quantitative population data are shown in Figure 5A–C (see Figure 5 legend for full description of treatment groups). As is the case for Brd2a knockdown [27], cell death is increased dramatically in the brains and dorsal spinal cords of prim5 morphants compared to NON-*brd2b*MO-injected or uninjected control embryos in every set (Figure 4A,E,F vs. Figure 4B,G; see Appendix A for HsRNA alone; Figure 5 A,B,C; 2bMO vs. control, HsRNA, or p53MO, respectively). The increased cell death in *Brd2*bMO morphant CNS is rescued substantially by co-injection of *HsBrd2*RNA, showing excess death to be a gene-specific effect (Figure 4B vs. Figure 4C; Figure 5A, 2bMO1 vs. 2bMO1 + HsRNA). Quantitative data show, on average, 707 apoptotic nuclei in the brains of *brd2b*MO morphants compared with 332 for uninjected and 334 for *brd2b*MO + HsRNA co-injected embryos (*p* < 0.0001, one-way ANOVA, Tukey’s HSD). Conversely, co-injection with *p53*MO does not substantially reduce the excess cell death observed (Figure 4G vs. Figure 4H; Figure 5C, 2bMO1 vs. 2bMO1 + p53MO), showing that excess death is not due to non-specific, off-target, p53-dependent effects sometimes observed with morpholino treatment [44]. Quantitative data show similar averages of apoptotic nuclei in the brains of *brd2b*MO-injected (581) and *brd2b*MO + *p53*MO co-injected (500) embryos, while uninjected and p53MO-injected embryos show, on average, between 306 and 288 apoptotic nuclei, respectively (*p* < 0.0001, one-way ANOVA, Tukey’s HSD). Finally, the dramatic increase in cell death seen in the CNS of morphants is recapitulated with Crispr-Cas9 disruption of the *brd2b* locus, supporting the claim that increased apoptosis is the result of Brd2b deficiency rather than general morpholino toxicity (Figure 4D,B,G; Figure 5B, 2bMO1, 2bMO2, Crispr2b vs. control). Quantitative data show, on average, 708 apoptotic nuclei in the brains of both *brd2b*MO1 and *brd2b*MO2 morphants and a similar average of 783 apoptotic nuclei in Crispr-disrupted embryos, while uninjected control embryos show, on average, 329 apoptotic nuclei (*p* < 0.0001, one-way ANOVA, Tukey’s HSD). Thus, deficiency during development of either Brd2a or Brd2b paralog results in dramatic increases in cell death in the CNS of segmentation stage embryos, likely accounting for a substantial part of the gross morphological defects observed in the developing brain.

Since Brd2b knockdown results in morphological defects of the ventral trunk, including a malformed and clogged pronephric duct and disorganized PBI, we looked in this region to assess cell death. As in Brd2a knockdowns [27], we consistently see excess cell death overall in the post-anal tail, PBI, and dorsal spinal cord of *brd2b*MO morphants (Figure 4B,D vs. Figure 4A,C; trunk insets, bottom arrows). Quantitative population data show, on average, 151 apoptotic nuclei in this region in *brd2b*MO morphants, compared to 75 in uninjected embryos and 86 in HsRNA-rescued morphants (*p* < 0.0001, one-way ANOVA, Tukey’s HSD). However, in terms of spatial distribution, we observe uniquely in *brd2b*MO morphants a *decrease* in cell death, sometimes along the ductal tube, but most consistently and prominently at the very distal cloaca, which normally would undergo cavitation via apoptosis to form the opening of the pronephric duct to the outside (Figure 4B,D vs. Figure 4A,C; trunk insets, top arrows). These observations suggest lack of cell death at the cloaca may be the cause of the plug of excess cells found there in Brd2b knockdowns, while increased overall cell death contributes to irregularities seen in the spinal cord, somites and PBI tissue of these morphants. Taken together, these findings show that as with Brd2a, paralog Brd2b is a major regulator of cell death in development and is capable of bimodal action—that is, it can act both as a pro-apoptotic and an anti-apoptotic factor, depending on context [27].

### 3.5. Patterning of pax2a (+) Spinal Interneurons and Distribution of Pronephric Cells Is Disrupted in Both brd2b and brd2a Morphants

In our previous study [27] we showed that zebrafish *brd2a* is necessary for the proper expression of some of the genes that pattern the MHB region (*eng2a* and *pax2a*) and the hindbrain (*krox20*/*egr2*). We also noted that *pax2a*-expressing spinal interneurons show abnormal patterning and numbers in *brd2a*MO morphants compared to control embryos, and thought it plausible that a HOX/Brd2a pathway might be involved. Normally, spinal interneurons derived from the ventral spinal cord are symmetrically paired on either side of the midline, and *homeobox* (*Hox*) genes are known to regulate their patterning and periodicity along the A–P axis [45]. To assess the effects of paralog Brd2b deficiency on these same parameters, we subjected a total of at least twenty prim 5 wildtype and *brd2b*MO morphant embryos per probe to hybridization In situ to *eng2a, pax2a*, and *krox20* for brain patterning defects and to *pax2a* for interneuron pairing defects (Figure 6A–L). In contrast to *brd2a*MO morphants, *brd2b*MO morphants show no differences in brain expression for the patterning genes tested (Figure 6A,D,G and Figure 6B,E,H vs. Figure 6C,F,I). However, when *pax2a*-expressing spinal interneurons were analyzed for matched pairing on either side of the nerve cord, we found a greater frequency of mispairing in *brd2b*MO morphants compared to uninjected or control-injected embryos (Figure 6J,K vs. Figure 6L), as observed in *brd2a*MO morphants [27]. In fact, quantitative population data obtained through anti-Pax2a immunofluorescence confocal imaging show, on average, 65.4% of spinal interneurons are unpaired in *brd2b*MO morphant embryos, compared to only 3.1% unpaired interneurons in controls (Figure 6M,N; *p* < 0.0001, chi square contingency, Fisher’s exact tests). This displays a similar trend to what we saw previously in *brd2a*MO morphants, which show unpaired interneurons 83% of the time, with 34.5% showing an unrecognizable pattern, compared to uninjected or control-injected embryos, which show, on average, mispairing only 3% of the time [27]. Thus, while patterning genes in the hindbrain and MHB region are misexpressed only in *brd2a* morphants, HOX-regulated patterning of spinal interneurons is disrupted by deficiencies in either paralog.

Since the developing pronephros appears malformed and the cloaca is plugged with excess cells uniquely in *brd2b*MO morphants, we wanted to examine the behavior of pronephric cells more closely. Pronephric progenitor cells normally express Pax2a, a key marker that distinguishes them from the blood cell progenitors that arise simultaneously from a shared field in the intermediate mesoderm [46]. Immunofluorescence staining followed by confocal imaging allowed us to assess the position and number of Pax2a(+) cells in individual segments of the developing pronephros (refer to Figure 6O for diagram) and compare them in prim 5 control embryos and *brd2b*MO morphants (Figure 6P,Q). In control embryos, Pax2a(+) presumptive pronephric cells are seen in the ventral trunk, lined up and evenly distributed along the length of the duct through the cloaca, where they form the opening to the outside (Figure 6P,Q; wt). In *brd2b*MO morphants, however, these same Pax2a(+) cells are unevenly distributed, with a paucity of cells in the middle segments of the duct, and an overabundance of cells occluding the cloaca at the very distal tip (Figure 6P, wt vs. *brd2b*MO1, arrows). Quantitative population data support these observations: *brd2b*MO morphants exhibit fewer Pax2a(+) cells in the subregion containing the proximal straight tube (PST), the distal early tube (DE), and the distal late tube (DL), but significantly more Pax2a(+) cells in the cloaca (C), compared to control embryos (Figure 6R, 2bMO vs. none; *p* < 0.0001, one-way ANOVA, Tukey’s HSD). No difference in cell number is seen between controls and *brd2b*MO morphants in the region of the glomerulus or the proximal convoluted tube (G/PCT) (Figure 6R, 2bMO vs. none; *p* = 0.3971, one-way ANOVA, Tukey’s HSD). As stated in the last section, cell death is especially—and consistently—reduced at the cloaca of the pronephros in *brd2b*MO morphants, and sometimes along the length of the duct. While reduced cell death may thus directly explain the excess cells at the cloaca, it cannot directly account for the *reduced* number of Pax2a(+) cells in midsections of the duct. It is possible that other mechanisms, such as reduced mitosis, misfating, or aberrant migration of pronephric precursors, might be at work in this region.

Strikingly, knockdown of paralog Brd2a results in almost opposite effects to those described above: there is a dramatic *excess* of Pax2a(+) presumptive pronephric progenitor cells along the entire length of the duct, but *not* at the distal tip, where the cloaca remains unclogged (Figure 6Q, wt vs. *brd2a*MO1, span between arrows). These data corroborate our previous In situ RNA hybridization studies, where *brd2a*MO morphants had too many *pax2>*a(+) cells to count in this region 76% of the time, whereas in uninjected or control injected embryos this occurred only 17% of the time [27]. Interestingly, cell death levels along the duct appear normal in these morphants, implying the involvement of other underlying mechanisms [47]. Overall, in the developing pronephros, paralogs *brd2a* and *brd2b* appear mostly to exert opposite effects on the number and distribution of Pax2a(+) presumptive pronephric progenitors during the period when the duct is being built.

### 3.6. Co-Knockdown of Brd2a and Brd2b Paralogs Restores Wild Type Phenotype to Both Brain and Pronephros of Morphant Embryos at Prim 5

Given that zebrafish paralogs *brd2a* and *brd2b* show substantially overlapping expression domains during embryogenesis, have the potential (at least theoretically) to form heterodimers via their respective bromodomains [48], and, when deficient, result in both similar and sometimes seemingly opposite defects, we wondered whether the paralogs interacted genetically. To test this, we performed co-injections of *brd2a*MO and *brd2b*MO to create double knockdowns and assessed the resultant morphant phenotypes. Single knockdowns were created by injecting 2–4 cell embryos with either 2 ng of *brd2b*MO or 4 ng of *brd2a*MO, the usual doses we use for each morpholino, respectively; double knockdowns were created by injecting a mix of *brd2b*MO + *brd2a*MO at the same individual concentrations. A total of 60 embryos per treatment were assessed at prim 5 stage for morphological defects. Representative images of embryos from each treatment group (wt, *brd2b*MO, *brd2a*MO, and *brd2b*MO+*brd2a*MO mix) are shown in Figure 7A–D; quantitative population data are shown in Table 1, *brd2a/brd2b co-knockdown*. Strikingly, co-injection of a mix of *brd2b*MO1+ *brd2a*MO1 suppresses the morphant brain and ventral trunk phenotypes and restores wild type morphology (Figure 7B,C vs. Figure 7A,D). This includes restoration of brain size, MHB region definition, organization of the PBI, and open pronephric cloaca. Quantitative population data support these observations (Table 1, *brd2a/brd2b co-knockdown*, *p* < 0.0001). We then used TUNEL assay to assess the effects of double knockdown on levels of cell death, and found that co-injection of morpholinos from the two loci restores wild type levels of apoptosis in both the CNS and ventral trunk of morphants—in particular, we see reduced cell death overall in the brains and trunks of co-knockdowns compared to either single gene morphant, but increased cell death at the cloaca of co-knockdowns compared to *brd2b*MO morphants (Figure 7F,G vs. Figure 7E,H). Quantitative analysis of apoptotic cell counts using optical sections through the brain of seven representative embryos from each treatment group indicates that single knockdown morphants display, on average, between 600 and 700 apoptotic nuclei, whereas co-injected embryos display, on average, around 300 apoptotic nuclei, comparable to uninjected embryos (Figure 5D; *p* < 0.0001, one-way ANOVA, Tukey’s HSD). When apoptosis was measured in the post-anal trunk in toto, including the pronephros, PBI, dorsal spinal cord and somites, double knockdowns show restoration to lower levels of cell death overall comparable to wild type (average 78 vs. 137–150 apoptotic nuclei, in double or uninjected vs. single knockdown morphants, respectively; *p* < 0.0001, one-way ANOVA, Tukey’s HSD). However, apoptosis increased in the cloaca in double-knockdowns compared to *brd2b*MO morphants, presumably opening the ductal exit (Figure 7F vs. Figure 7H, trunk, white arrow in Figure 7F). Double knockdown also corrects the aberrant distribution of Pax2a(+) cells along the length of the duct and at the cloaca observed in single knockdowns (Figure 7I–L; cloaca, white arrow in left panels; ductal tube, span between white arrows in right panels). Whereas *brd2b*MO morphants show clogged cloaca and sparse distribution of Pax2a(+) cells along the mid-sections of the ductal tube (Figure 7J), and *brd2a*MO morphants show open cloaca and excess Pax2a(+) cells along the ductal tube (Figure 7K), both wildtype and double knockdown embryos show open cloaca and more even distribution of Pax2a(+) pronephric cells along the duct (Figure 7I,L). Finally, double knockdown suppresses the mispairing defect of VNC-derived spinal interneurons seen in each single knockdown, increasing the number and pairing of interneurons to near wildtype levels (Figure 7M,P vs. Figure 7N,O; pairs indicated by lines in each panel).

Thus, in the double knockdown, the deficiency phenotypes of each paralog are suppressed by the concurrent deficiency of the other paralog. Taken together, these results suggest some sort of genetic interaction, direct or indirect, with functional antagonism, is at work between the two paralogs *brd2b* and *brd2a* during development of the CNS, pronephros, and the PBI in zebrafish embryos.

### 3.7. Enhancement of Morphant Brain Phenotypes by Injection of Paralogous RNA Corroborates Genetic Antagonism between brd2a and brd2b Loci

Given that Brd2b/Brd2a double knockdown restores wild type phenotype, we reasoned that the proposed genetic interaction between the two paralogs might involve functional antagonism and/or some required stoichiometry between their respective gene products. To explore this idea, we conducted a series of genetic rescue and enhancement studies using combinations of morpholinos and in vitro-synthesized mRNAs aimed at testing the following predictions: (1) *brd2b*MO morphant phenotypes will be *rescued* by co-injection of cognate *brd2b* RNA but *enhanced* by co-injection of paralogous *brd2a* RNA; and conversely, (2) *brd2a*MO morphant phenotypes will be *rescued* by co-injection of *brd2a* RNA but *enhanced* by co-injection of paralogous *brd2b* RNA. Up to this point, we had used in vitro-synthesized human *Brd2* RNA in our rescue experiments to verify the specificity of the original *brd2a* and *brd2b* morpholino knockdowns. In fact, the human transcript was able to rescue to a notable degree the knockdown phenotypes of both *Brd2* paralogs in zebrafish [27]. However, to conduct this next set of rescue/enhancement studies, we first cloned full-length cDNAs of both *brd2a* and *brd2b* using RT-PCR on mRNA isolated from zebrafish embryos (see Materials and Methods). We obtained a single 3.8 kb cDNA representing the major full-length transcript of *brd2a*, and two cDNAs of 3.5 kb and 1.4 kb, representing long (*brd2b*-L) and short (*brd2b*-S) transcript variants, respectively, from the *brd2b* locus. Thus, we could conduct rescue-enhancement studies using all three transcripts derived from native zebrafish genomic loci.

To test the newly cloned zebrafish cDNAs, we first set up rescue experiments for each paralog using full-length RNAs. Treatment groups for *brd2b*MO rescue (control uninjected, *brd2b*MO2, *brd2b-L*RNA, and *brd2b*MO2 + *brd2b-L*RNA co-injected), and for *brd2a*MO rescue (control uninjected, *brd2a*MO1, *brd2a*RNA, and *Brd2*MO1 + *brd2a*RNA co-injected), were assessed for brain and trunk morphology. For quantitative population analysis of all rescue-enhancement experiments, we counted only defects that could be classified as moderate to severe, to avoid any ambiguity in scoring between wild type embryos and mildly defective morphants. Representative images from these experiments are shown in Figure 8 and quantitative population data are presented in Table 2. *brd2b*MO morphant brain defects are partially rescued by co-injection of zebrafish *brd2b-L*RNA (Figure 8D vs. Figure 8E; arrow), with some restoration of the rectangular dorsal width and height of the MHB region and clearer delineation of the boundary itself. Quantitative population data support brain rescue: *brd2b*MO2-injected morphants present with moderate to severe brain defects 40.4% of the time, while *Brd2*MO2+*brd2b-L*RNA co-injected morphants present moderate/severe defects only 24.6% of the time (Table 2, *2bMO rescue 2bRNA-L_brain*; *p* = 0.0026). Correspondence analysis clusters wild type phenotype with [control, 2bMO+2bRNA-L] treatments and defective phenotype with [2bMO, 2bRNA-L] treatments. Similarly, *brd2a*MO morphant brain defects are partially rescued by co-injection of zebrafish *brd2a*RNA (Figure 8G vs. Figure 8H; arrow), with height and width of MHB region and size of the ventricle and hindbrain partially restored. Quantitative population data support brain rescue: *brd2a*MO morphants show moderate to severe brain defects 35% of the time, compared to 20% for embryos co-injected with *brd2a*RNA (Table 2, *2aMO rescue 2aRNA_brain*; *p* = 0.1416). Although above threshold for statistically significant differences among treatments, Correspondence analysis clusters wild type with [control] then [2aRNA, 2aMO+2aRNA] treatments, while defective phenotype is grouped more definitively with [2aMO] treatment. Thus, knockdowns of each paralogous locus are at least partially rescued in terms of brain defects by introduction of their full-length, cognate zebrafish RNAs, in a fashion similar to their collective rescue by the single human *Brd2* RNA [27]. We noted that injection of either zebrafish *brd2b-L*RNA or *brd2a*RNA alone into embryos often results in mild brain defects that phenocopy morphant defects (Figure 8, compare Figure 8B and Figure 8D; compare Figure 8C and Figure 8G; see also Table 2, *2bMO rescue-2bRNA-L_brain* and *2aMO rescue-2aRNA_brain*, compare RNA injections vs. control). We believe this, in fact, supports the idea of a required stoichiometry between zebrafish paralogs, as the injection of native RNAs from either locus would be predicted to skew that ratio, as would a deficiency of either paralog, This differs from injection of human *Brd2* mRNA alone, which at this same concentration, has no effect [27], suggesting that adding human *Brd2* RNA may not disturb this ratio, as it may encode a protein that carries both Brd2a and Brd2b activity.

In contrast to the brain, the morphology of the pronephric duct in embryos of *brd2b*MO rescue studies shows no improvement using *brd2b-L*RNA co-injections, with Correspondence analysis clustering wild type with [control, 2bRNA-L] and defect with [2bMO, 2bMO + 2bRNA-L] (Table 2, *2bMO rescue 2bRNA-L_duct*, *p* = 0.0002). Intriguingly, robust rescue of the duct does occur when we co-inject the short transcript variant *brd2b-S*RNA instead: co-injection of *brd2b-S*RNA reduces the incidence of defects to 29.6% compared to 80% for *brd2b*MO morphants, with Correspondence analysis grouping wildtype with [control, 2bMO+2bRNA-S] and defect with [2bRNA-S, 2bMO] (Table 2, *2bMO rescue 2bRNA-S_duct*, *p* = 0.0128). The short *brd2b-S*RNA transcript does not, however, rescue *brd2b*MO morphant brain defects (Table 2, *2bMO rescue 2bRNA-S_brain*, *p* = 0.0840), suggesting that *brd2b-S*RNA has specific functions in the developing pronephros but is less critical for normal brain formation.

Once we verified that zebrafish *Brd2* RNAs from each locus could rescue their respective MO knockdown phenotypes, we used these same RNAs in corresponding genetic enhancement studies (see Materials and Methods). Treatment groups for *brd2b*MO enhancement (control uninjected, *brd2b*MO2, *brd2a*RNA, and *brd2b*MO2 + *brd2a*RNA co-injection) and for *brd2a*MO enhancement (control uninjected, *brd2a*MO1, *brd2b-L*RNA, and *brd2a*MO1 + *brd2b-L*RNA co-injection), were assessed for brain and trunk morphology. Representative brain images from these experiments are shown in Figure 8 and quantitative population data are presented in Table 2. *brd2b*MO morphant brain defects are often enhanced by co-injection of paralogous *brd2a*RNA (Figure 8D vs. Figure 8F), with a smaller brain and a more triangular-shaped MBH region (arrow in Figure 8F). Similarly, *brd2a*MO morphant brain defects are often enhanced by co-injection of paralogous *brd2b*RNA-L (Figure 8G vs. Figure 8I), with further loss of brain segment definition (arrow in Figure 8I). Quantitative analysis of population data support these brain enhancement observations in terms of penetrance: *brd2b*MO moderate brain defects increase in incidence from 30.8 to 42.9%, and *brd2a*MO brain defects increase from 41.5 to 61.4%, with co-injection of their respective full-length paralogous RNAs (Table 2, *2bMO enhancement 2aRNA_brain*, *p* = 0.0059; *2aMO enhancement 2bRNA-L_ brain*, *p* = 0.0109). In addition, Correspondence analysis groups wild type with [control], and moderate defect most closely with [2bMO + 2aRNA], in each case. In contrast, the short transcript, *brd2b-S*, does not enhance the *brd2a*MO brain defect (Table 2, *2aMO enhancement-2b-S_brain*, *p* < 0.0001), with Correspondence analysis grouping defect closely with both [2aMO, 2aMO+2bRNA-S] As for the pronephros, co-injection of *brd2a*RNA enhances the *brd2b*MO morphant phenotype in the duct also, increasing the incidence of defects over that seen with either reagent alone (Table 2, *2bMO enhancement 2aRNA_duct*, *p* < 0.0001). Correspondence analysis groups wild type with [control] followed by [2aRNA, 2bMO1], with defect most closely associated with [2bMO1 + 2aRNA]. Thus, levels of both *brd2b-S* RNA (as seen in rescue above) and *brd2a* RNA (as seen in enhancement here) may be involved in negotiating normal pronephric duct development. Taken together, these studies confirm that enhancement with increased penetrance of *brd2a*MO and *brd2b*MO morphant brain and duct phenotypes is obtained by co-injection of the paralogous full-length RNA, which presumably would further skew the Brd2a/Brd2b ratio.

To corroborate our morphological observations, we extended genetic rescue-enhancement studies to assess differential cell death levels in prim 5 individuals across treatment groups using TUNEL assays. Although the previous rescue-enhancement treatments could be evaluated in terms both of penetrance and expressivity of brain defects, this was less true with respect to the pronephric duct, where we saw penetrance differences but were unable to parse out severity levels. Consequently, we focused on brain samples for quantitative studies of differential cell death levels that might underlie morphologies observed. Treatment groups for cell death rescue-enhancement studies were the same as for morphological experiments. Representative samples for each treatment are shown in Figure 8J–R; quantitative population data are shown in Figure 5E–H. Consistent with morphological observations, elevated levels of cell death in morphants were partly rescued by co-injection of RNA from the corresponding locus (Figure 8M vs. Figure 8N; Figure 8P vs. Figure 8Q). Quantitative population data support these observations: *brd2b*MO morphant brains displayed, on average, 534.67 apoptotic nuclei compared to 367.33 in embryos co-injected with cognate *brd2b-L*RNA (Figure 5E, *p* = 0.0286); furthermore, *brd2a*MO morphant brains showed, on average, 848.0 apoptotic nuclei compared to 562.7 in embryos co-injected with cognate *brd2a*RNA (Figure 5G, *p* = 0.0176). Conversely, co-injection with the RNA of the paralogous locus exacerbated the excess cell death in morphants (Figure 8M vs. Figure 8O; Figure 8P vs. Figure 8R). Quantitative population data are consistent with these observations: *brd2b*MO morphants displayed, on average, 457.0 apoptotic nuclei compared to 806.0 in embryos co-injected with paralogous *brd2a*RNA (Figure 5F, *p* = 0.0008); similarly, *brd2a*MO morphants showed an increase from 743.7 to 1167.0 apoptotic nuclei in embryos co-injected with paralogous *brd2b-L*RNA (Figure 5H, *p* = 0.0084). As noted in morphological studies, injection of either *brd2a*RNA or *brd2b-L*RNA alone results in a mild phenocopy, with cell death levels elevated above wild type, but not as high as in MO-treated morphants (Figure 8J vs. Figure 8K,L; Figure 5E–H, compare none and RNA alone injections in each case), again supporting the idea of a required stoichiometry between paralogs. Taken together, these results suggest that paralogs *brd2b* and *brd2a *interact genetically as functional antagonists during zebrafish development to effect proper formation of the brain, in part through their co-regulation of cell death.

## 4. Discussion

Our comparative studies of zebrafish *brd2a* and *brd2b* reveal both shared and unique structural, regulatory, and functional features in the two paralogs, and provide a concrete example of how gene duplication and divergence might contribute to the evolution of gene networks and developmental pathways. In the case of zebrafish *Brd2*, diversification is exemplified through paralog-specific RNA variants and protein isoforms, distinct localization of maternal factors, shared and unique tissue- and stage-specific expression and function, and novel genetic interactions.

### 4.1. Shared and Distinct Functions of Brd2b/2a Paralogs

#### 4.1.1. RNA Variants and Protein Isoforms

Both *Brd2* paralogs exhibit RNA variants encoding putative protein isoforms lacking one or more of the canonical BET domains. We show that long and short *brd2b* transcript variants are distinctly regulated: whereas *brd2b-L *RNA is a maternal-zygotic transcript expressed in both oocytes and embryos, *brd2b-S* is a zygotic transcript expressed specifically in segmentation stage and older embryos. The short transcript encodes a predicted protein that carries only bromodomain 1 (BD1); this might function as a dominant-negative factor by binding target loci of full-length Brd2b or Brd2a but failing to bring associated protein cargo. This idea is supported by recent studies showing the ET domain to be crucial for Brd2 association with chromatin and for its transcriptional and splicing effects [49]. Alternatively, or in addition, Brd2b-S protein (or the *brd2b*-S RNA itself) could have its own unique co-transcriptional or other activity. In any case, the short *brd2b* variant/isoform specifically is critical to proper formation of the pronephric duct, consistent with its zygotic expression during segmentation. A third predicted *brd2b* transcript variant that is listed in the Ensemble database (GRCz11 release) but is as-yet unsupported by cDNA evidence, encodes a different truncated protein containing a partial BD2, an NLS and ET domain (brd2b-201 from ENSDARG_00000046087). Similarly, paralog *brd2a* produces two verified transcript variants encoding nearly identical full-length BET proteins [27] (NCBI NM_001270571.1, NM_131200.2), and two predicted transcripts encoding truncated proteins—one containing BD1 and partial BD2 alone (NCBI XM_021467982.1), and another, a partial BD1 alone (Ensemble, GRCz11 release, brd2a-202 from ENSDARG_00000022280), respectively. Thus, transcript variants and protein isoforms with different subsets of BET domains may be characteristic of *Brd2* paralogs in zebrafish, and possibly, of *Brd2* and other BET genes in other species. In fact, according to the NCBI Gene database, the human *Brd2* locus produces up to five RNA variants resulting in four predicted protein isoforms, one of which lacks the majority of BD1 (NP_001278915.1). Other potential isoforms lacking the ET domain or containing only BD1 (as in Brd2b-S here) are listed in Ensemble (GRCh38; ENSG00000204256.14). Interestingly, both databases also list non-coding RNAs or RNAs with retained introns for human *Brd2*. Long and short transcripts from alternate promoters, and alternatively spliced transcripts retaining part of a conserved intron carrying a premature stop codon (such as *brd2b-S*), were previously observed for mammalian *Brd2* [50]; intriguingly, the ratio of regular to alternative splice variants is regulated by the length of a polymorphic repeat nearby, implying stoichiometry among variants is key [50]. Finally, in flies, zebrafish, mice and humans, *Brd2* RNA variants and/or protein isoforms may be preferentially expressed in distinct tissues or stages [4,27,50,51]. Together, these observations suggest complex modularity among Brd2 isoforms, and possibly combinatorial regulatory outcomes from their division of labor and/or interactions.

#### 4.1.2. Maternal-Zygotic Function

All known *Brd2* species homologs, as well as both *Brd2* paralogs in zebrafish, encode maternal and zygotic factors, implying conserved function in germline and early developmental events [23,24,25]. However, while zygotic factors from the two zebrafish paralogs overlap greatly in spatial expression in embryos, maternal factors localize to strikingly different regions in stage IV oocytes—Brd2a in the peripheral cortical cytoplasm and Brd2b restricted to the future animal pole ([25,27], and this study). This suggests functional divergence between paralogs, perhaps in epigenetic reprogramming events at the earliest stages of development. In fact, epigenomic studies reveal unique chromatin landscapes in oocytes, sperm, and early embryos, including differences in chromosome organization and accessibility [52]. Significantly, the epigenetics and 3D architectures of the genome undergo dramatic changes at developmental transitions such as those associated with egg maturation, meiotic reentry, the egg-to-embryo transition, and activation of zygotic transcription [53]. Presumably, this leads to selective access to chromatin and correct deployment of gene regulatory programs. By localizing to distinct regions within oocytes, zebrafish Brd2 paralogs may access different gene sets or organize chromosome topologies in different areas of the genome, in preparation for embryogenesis. Recently, BET proteins have been directly implicated in higher order chromatin structure and chromosome organization as well as in regulation of developmental transcriptional programs. For example, testis-specific BRDT regulates chromatin organization and epigenetic reprogramming during prophase I of male mammalian gametes and is necessary for the defined topology of the chromocenter in post-meiotic spermatids [8]. In embryonic stem cells (ESCs), Brd2 acts as an antagonist of histone 2A.Z (H2A.Z) at promoters of poised, bivalent genes, maintaining a balance between activation and repression of chromatin that is critical to proper execution of developmental programs [54]. Finally, and more generally, Brd2 co-localizes with chromatin insulator protein CTCF genome-wide to effect transcriptional boundaries between sets of genes, a “bookmarking” activity that potentiates gene co-regulation [55].

The zygotic activities of both *Brd2* paralogs center on the formation and patterning of segmented or compartmentalized tissues including the brain, spinal interneurons, PBI, and pronephric duct, echoing ancient Hox-dependent segmentation functions from Drosophila [4,23]. Indeed, the fly homolog of *Brd2*, *fs(1)h*, encodes long and short isoforms, and exhibits maternal and zygotic mutant alleles that affect different BET domains and cause distinct segmentation and homeotic defects. For instance, maternal alleles carrying a deletion of the ET domain of the long isoform lead to segmentation defects via *homeobox * gene *engrailed (eng)* misexpression and trunk homeosis via the *homeobox* gene *Ultrabithorax* (*Ubx*), whereas a mutation in BD-1 in this same isoform leads to termini deletions via RAS signaling [26]. In contrast, a zygotic allele carrying a BD-1 mutation in the short isoform leads to head homeosis via misexpression of *homeobox* genes *Deformed (Dfd)* and *labial (lab)*, and to gut defects via *cap’n’collar* (*cnc*) [26]. Interestingly, another zygotic temperature-sensitive allele of *fsh-S* results in reduced *Ubx* expression in the VNC of Drosophila embryos [4], suggesting that a Brd2/Hox pathway might be involved in VNC-derived interneuron mispairing seen in both zebrafish *brd2a *and *brd2b* morphants [25,27]. Recent work has uncovered a role for *Hox* genes in morphogenesis and lineage fidelity in developing mammalian kidney, suggesting that pronephric abnormalities seen in *brd2a* and *brd2b *morphants may also involve misregulation of *homeobox* genes [56]. Thus, modularity, temporal-spatial specificity of isoforms, distinct maternal and zygotic functions, and intersection with Hox-dependent patterning pathways, are ancient features of Brd2 that appear to be retained across species and in both zebrafish paralogs. Zebrafish paralogs appear to diverge, however, in some downstream patterning genes in the brain such as *eng2a*, *krox20*, and *pax2a*, where only *brd2a* deficiency results in expression defects [27]. Thus, *Brd2* paralogs may act in both shared and distinct processes during CNS development, perhaps displaying redundancy or sub-specialization depending on context.

#### 4.1.3. Cell Death, Differentiation, and Lineage Specification

Both *Brd2* paralogs affect the developing CNS, PBI, and pronephric duct in large part by suppressing cell death during stages when differentiation, patterning, morphogenesis, and lineage specification are ongoing. In the CNS, the deficiency of either paralog results in nearly identical defects: high levels of neuronal apoptosis leading to reduced hindbrain, an ill-defined MHB region, and a deformed spinal cord. There is growing evidence of the importance of Brd2 across species in the control of neuronal apoptosis, differentiation, and/or proliferation. Similar in many ways to zebrafish knockdowns, Brd2 knockout mice exhibit excess apoptosis, reduced brain, neurulation defects, hindbrain exencephaly and transcriptome changes in neurogenesis genes, prior to embryonic lethality [31,32]. Other studies show Brd2 is necessary for the cell cycle exit and neuronal differentiation of mouse neuroepithelial cells, as a deficiency of Brd2 increases proliferation and decreases neuronal differentiation [36]. Significantly, this phenotype is suppressed in Brd2/E2F double mutants, implying an antagonistic relation between Brd2 and cell cycle promoter E2F in this neuronal context. Finally, in a Brd2 (+/−) haploinsufficient mouse model of juvenile monoclonal epilepsy (JME), the number of GABAergic neurons is reduced in the striatum and primary motor cortex, suggesting that Brd2-dependent neurogenesis is critical for the proper differentiation of specific neuronal cell types [38]. A connection between Brd2 and neuronal differentiation is also seen in Drosophila, as a zygotic loss of function (LOF) allele of *fsh-S* results in loss of terminal dendritic branches and impaired mechanosensory responses [57]. Together, these observations suggest that the involvement of Brd2 in neuronal development is ancient and evolutionarily conserved, although a more recent role for Brd2, specifically in neuronal apoptosis, only, as far as we know, appears in vertebrates, and is retained in both zebrafish paralogs.

Both zebrafish *Brd2* paralogs are likely involved in the formation of blood and pronephric (kidney) tissues, but display some unique, even opposing, phenotypes when deficient. Given that blood and kidney precursors arise from a common progenitor field in the intermediate mesoderm [46], paralog activity might affect lineage specification and/or concurrent processes of mitosis, apoptosis, and cellular differentiation. The deficiency of either paralog results in an unstructured and sparsely populated PBI, likely due in part to the increased apoptosis in the region. However, the penetrance of this defect differs greatly between paralogs, implicating *brd2a* as the prevailing regulator in PBI tissue. In contrast, the two paralogs appear to act in opposite directions when it comes to the pronephric duct: whereas knockdown of *brd2a* results in a dramatic increase, knockdown of *brd2b* shows a significant decrease, of Pax2a(+) pronephric cells along the ductal tube. Misregulation of apoptosis is unlikely to account for either an excess or deficit of ductal tube cells, as levels of cell death and numbers of cells in this region do not co-vary in the expected directions in either *Brd2*MO morphant. This leads us to believe that differential mitosis or differentiation and/or a switch in cell fate specification may be involved. Brd2a deficient embryos may display an increase in kidney precursors at the expense of blood precursors, for instance [47], which would be consistent with the known requirement of Brd2 for proper hematopoiesis in mammals [13,30] and zebrafish [29]. A careful examination of knockdown effects in this region using cell-type-specific markers for blood and kidney lineages would help elucidate mechanisms at play. Misregulation of apoptosis is likely involved specifically at the cloaca, however, as *lack* of apoptosis in *brd2b*MO morphants correlates with *excess* cells plugging the opening. It is also possible that a change in distribution or migration of Pax2a(+) cells to the distal tip accounts for the cloacal defect in *brd2b*MO morphants, as we see an increase in cells at the cloaca with a concomitant decrease in cells along the tube. Taken together, these findings suggest that, normally, *brd2a* may promote the formation of blood precursors, while *brd2b* aids in the proper formation/distribution of kidney precursors, and furthermore, that paralogs work together to achieve a normal PBI and duct, possibly through their combined regulation of cell death, mitosis, and/or (possibly Hox-dependent) differentiation during lineage specification in the ventral trunk. More generally, the two paralogs appear capable of exerting opposite phenotypic outcomes, depending on context and tissue.

Corroborating these findings, work using genetics and small molecule BET inhibitors provide evidence for the involvement of Brd2 and/or other BET proteins in hematopoiesis [13,29,30], erythroid specification and transcriptome activation [13], hematological malignancies [17,18,30,34], and the modulation of pathways relevant to chronic kidney disease [19]. In addition, BET proteins are implicated as regulators of cell state transitions and cellular identity in various models [10,11,12,13,14]. Significantly, basic helix-loop-helix (bHLH) transcription factors mediate the epigenetic response elicited by the treatment of some cancer cells with BET inhibitors, implying that BET proteins connect specifically with transcriptional networks of bHLH-influenced developmental pathways—namely, those that critically regulate neurogenesis, cell proliferation and differentiation, and cell lineage determination [58]. Finally, lineage-specific transcription factors are known to recruit BET proteins to specific chromatin sites in response to cellular signals, inducing transcriptional programs and stabilizing desired cell states. For example, recruitment of Brd2 by transcription factor Lyar helps reduce levels of pro-proliferative Nanog upon differentiation signaling; deletion of the Brd2-binding site of Lyar leads to impaired differentiation and increased cell death [59]. In another context, an isoform switch between Brd2 and Brd4 occupancy at Nodal-regulatory elements (NRE) dictates whether cells retain the pluripotent state or take on a lineage-specific fate [60]. Brd2 is also involved in mediating fate switching in osteoclast [61] and erythroid [62] lineages, and in the epithelial to mesenchymal transition (EMT) [63], where it acts antagonistically with Brd3/4. Possibly, a similar switching mechanism between Brd2a and Brd2b paralogs creates the proper balance of Pax2a(+) kidney vs. Pax2a(−) blood precursors from a common progenitor field.

### 4.2. Functional Antagonism between Brd2 Paralogs

We provide evidence for genetic interaction with functional antagonism between *brd2a* and *brd2b*, showing that double knockdown suppresses the morphant defects we observe when either paralog is knocked down individually. Strikingly, double knockdown suppression rescues defects whether they are similar or opposite in effect in each single knockdown, or in fact, unique to a single knockdown. Thus, antagonism between paralogs may work in multiple contexts and via shared and individual regulatory pathways. We provide support for the idea that a proper ratio between the two paralogs must exist at some level for normal development to occur, by further skewing that presumed ratio with co-injection of paralogous RNA, and thereby enhancing single knockdown defects. Whether the balance of gene products is achieved through direct or indirect interaction between paralogs is yet unknown. Both homodimers and heterodimers can form among BET family members [1,48], so direct interaction is plausible; indeed, a conserved motif B in bromodomains provides a surface for homo- and hetero-dimerization, and is necessary for association with chromatin [48]. In any case, a skewed 2a:2b ratio presumably could alter the concentration-dependent regulation of specific developmental pathways, leading to, for instance, excess cell death in the CNS or a switch in cell fate in the ventral trunk, ultimately explaining the defects we observe. A summary of our findings and a potential model for interaction with functional antagonism between paralogs, assuming direct hetero- and homodimer formation, is presented in Table 3, and may be referred to in the discussion below.

Our study of *brd2a* and *brd2b* paralogs in different tissues suggests a variety of possible mechanisms for functional antagonism. In the brain, a deficiency in either Brd2a or Brd2b results in the same defects of excess apoptosis and dysmorphology, implying similar action, occurring perhaps in a shared pathway. These defects are suppressed by double knockdown, enhanced by co-injection of each morpholino with the paralogous RNA, and phenocopied by injection of the respective paralogous RNA alone. This suggests a functional antagonism based on a proper balance of 2a:2b, where skewing in either direction results in pathway misregulation and ectopic cell death (Table 3, CNS). Alternatively, the paralogs may participate in separate pathways that intersect at some point down the line. Enhancement and phenocopying of *brd2a*MO occur specifically with *brdb2-L* rather than *brd2b-S* RNA injections, implicating full-length Brd2b-L as co-regulator with Brd2a of apoptosis in the developing CNS.

In the ventral trunk, the morphogenesis of the PBI and pronephric duct is affected by deficiencies in either paralog, but in diverse ways. As in the brain, a deficiency in either paralog results in excess cell death in the PBI, and this defect is substantially suppressed by double knockdown. Thus again, a balance between the two paralogs may be required for normal PBI formation, and a relative deficit of one or the other leads to excess apoptosis in this region (Table 3, PBI). The PBI is considered a transitional hematopoietic tissue where erythromyeloid progenitors (EMPs) mature into erythroid and myeloid cells from 19 to 24 hpf [43]. As these various cell types coexist in the PBI at the time of our assay (24 hpf), it is possible that the paralogs act in different lineages, with some crosstalk between them to allow for double knockdown suppression.

As mentioned already, *Brd2* paralogs exhibit opposite morphant defects in the developing pronephric duct: while Brd2a deficiency results in an excess of Pax2a(+) pronephric cells along the ductal tube, a deficiency of Brd2b results in fewer such cells. These single knockdown defects are partially suppressed in the double knockdown, such that the cell density and distribution are closer to wildtype. As both pronephric and blood cells normally reside in this region of the ventral trunk, several mechanisms of antagonism are possible in which Brd2a promotes blood cell fate, proliferation and/or survival, while Brd2b does the same in pronephric cells. Thus, in the ductal tube region, the 2a:2b ratio might function as a concentration-dependent switch, where skewing in either direction upsets a balance between alternate developmental programs and/or fates in blood and kidney cells (Table 3, Pronephric duct). A unique situation occurs at the distal tip of the pronephric duct, where a deficiency of Brd2b—but not of Brd2a—results in a plug of excess Pax2a(+) cells at the cloaca with concomitant reduced apoptosis. Cloacal defects are substantially suppressed in the double knockdown, enhanced by co-injection of paralogous *brd2a* RNA, and phenocopied by ectopic injection of paralogous *brd2a* RNA alone. This suggests an antagonistic ratio that is allowed to skew in only one direction: levels of Brd2b must surpass a minimum threshold, while levels of Brd2a must fall below maximum threshold (Table 3, Cloaca). Reduced Brd2b might still be compensated by reduced antagonist Brd2a in the double knockdown. Significantly, cloacal occlusion is rescued selectively by *brd2b-S* rather than *brd2b-L* RNA; so the truncated Brd2b-S isoform, and presumably, the Brd2b-S:Brd2a ratio, appear important for proper formation of this structure. The onset of expression of *brd2b-S* RNA by early segmentation and its perdurance until at least 48 hpf correlates with the timing of pronephric duct morphogenesis, and is consistent with these data. Intriguingly, *brd2a* transcripts appear transiently degraded in vivo during mid-segmentation in wildtype when *brd2b-S* is most prominent [27], skewing the ratio in the proposed “permitted” direction. A complicating factor is that injection of *brd2b-S* RNA alone also phenocopies the *brd2b*MO defect at the cloaca—this might indicate a ceiling to *brd2b-S* expression, above which interactions with other targets may occur that interfere with the normal course of events at the cloaca. The fact that Brd2b-S is missing the ET domain and may act as a dominant negative, might allow it to repress BET proteins other than Brd2a, for instance.

Taken together, these studies show that *Brd2* paralogs in zebrafish have developed novel interactions that work in concentration-dependent and antagonistic ways to modulate shared, one-sided, and/or alternate developmental pathways in different tissues and developmental contexts.

### 4.3. Gene Duplication, Divergence, and Antagonism

Numerous studies describe paralogous genes that have developed interactions and/or antagonistic relationships of various kinds, highlighting the evolutionary significance of such events, and providing precedent and context for our findings. First, BET proteins are known to interact to effect different regulatory outcomes. For example, Brd2 and Brd4 co-occupy superenhancer elements to regulate the inflammatory response [6], and both cooperate with Myc at shared sites to promote osteoclast over macrophage cell fate [61]. In addition, Brd2, 3, and 4 are recruited to GATA-regulated genes, in a preferred stoichiometry, to activate erythroid transcription; in this context, ectopic Brd3 rescues Brd2, but not Brd4, deficiency, implying both distinct and redundant BET functions [62]. On the other hand, Brd2 and Brd3/4 activate independent transcriptional networks and thereby compete with one another to promote or repress, respectively, the epithelial to mesenchymal transition (EMT) in a breast cancer model [63]. Thus, BET proteins interact both cooperatively and antagonistically, and in both indirect genetic and direct biochemical fashion. Second, other more recently diverged paralogs are known to interact in ways that resemble what we see with zebrafish *Brd2*. For instance, antagonistic paralogs can affect a single pathway in opposite direction, as do Ubf3a and 3b in the nonsense-mediated decay (NMD) pathway [64]. Here, repressor paralog Upf3a lacks a critical domain and acts as a dominant negative “rheostat” on active paralog Upf3b by competitive blocking. Dominant-negative mechanisms are considered common in gene evolution [64] and may be applicable in our case to Brd2-S. Antagonistic paralogs can also act as ON/OFF switches of a single pathway by having opposite transcriptional effects on the same set of target genes, as pals-22 and pals-25 do in the intracellular pathogen response (IPR) pathway of C. elegans [65]. On the other hand, paralogs can control different unique pathways by their concentration-dependent effects on a common molecular target. For example, paralogs antagonistically controlling epigenetic co-regulator lysine demethylase 1 (LSD-1) during hematopoiesis, either promote megakaryocyte cell fate or lead to erythroid differentiation from a common precursor, depending on their relative concentrations [66]. Something along these lines might characterize the action of zebrafish *Brd2* paralogs in blood versus pronephric lineages. Finally, paralogs often exhibit sub-functionalization after duplication, presumably allowing division of labor and complementary interactions [67]. We believe this likely occurred here, as the single human *Brd2* RNA rescues morphant defects of either zebrafish paralog, while zebrafish RNA from each paralogous locus rescues only its own morphant defects. In addition, *brd2a* and *brd2b* exhibit at least some unique spatiotemporal expression patterns, both in oocytes and embryos [25], suggesting functions that may be completely independent and the result of either sub-functionalization or neofunctionalization. It is also possible, however, that Brd2b is simply a repressor of Brd2a, and has no other independent function of its own. In this case, the levels of Brd2a would be all that matter, causing defects when either too high or too low, depending on context. We believe this might be true of truncated Brd2b-S, but less likely for full-length Brd2b-L, given the history of BET protein *interse* interactions. Finding a defect unique to one paralog that is not suppressed by the deficiency of the other paralog, and/or attempting rescue of nulls of either paralog with ectopic expression of the other, might help to definitively resolve this issue.

## 5. Conclusions

In summary, our study of zebrafish *Brd2* paralogs provides a concrete example of divergence after gene duplication that illustrates multiple levels of diversification, including the following: paralog-specific variants/isoforms and domain configuration changes; both shared and unique spatiotemporal expression and function; both shared and possibly unique developmental pathways; and concentration-dependent genetic interaction with functional antagonism. It will be of future interest to assess paralogs for direct biochemical interaction, to identify their shared and unique upstream regulators and downstream targets, and to compare transcriptome changes brought about by single and double knockdowns. Diversification of paralogs has allowed for increased complexity of regulatory networks, more crosstalk between networks controlling different processes, and greater selectivity and specificity of signaling pathways [68]. Thus, studies such as these deepen our understanding of a major mechanism of evolution and development. Just as importantly, the study of *Brd2* paralogs in zebrafish may provide novel insights into possible functions, activities, and interactions of human Brd2, which as a member of the BET family of epigenetic co-regulators, stands at the intersection of multiple regulatory pathways, including those involved in carcinogenesis and neurodegeneration. Master regulatory proteins such as Brd2 are likely to belong to “critical paralog groups”—the subset of paralogs that are more central to signaling pathways, more varied in biological function and complex in post-transcriptional regulation, and more likely to be mutated in genetic diseases and cancer [69]. Their investigation presents a unique opportunity for understanding the mechanisms of control circuits critical for development and normal biological function, as well as their evolution and complexification over time.

## Figures and Tables

**Figure 1 jdb-09-00046-f001:**
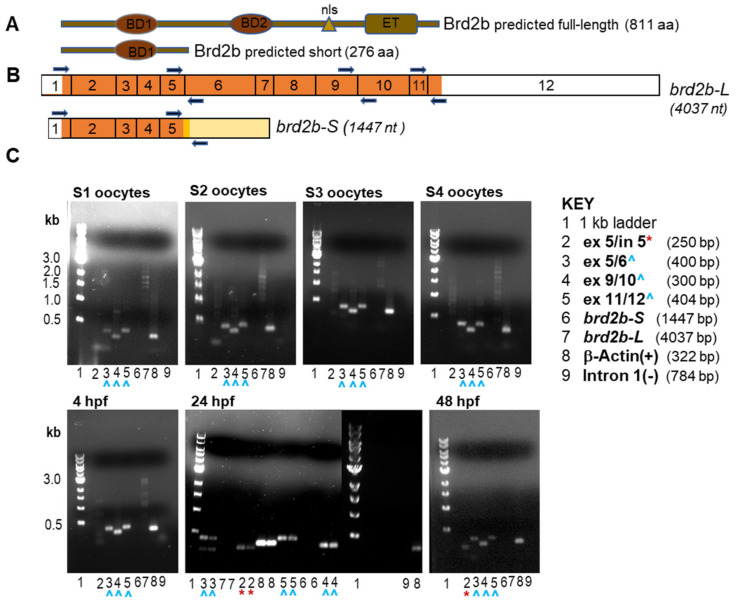
*brd2b* transcript variants are differentially regulated during development. (**A**) Schematic of predicted proteins from the *brd2b *locus, full-length Brd2b-L and truncated Brd2b-S. Bromodomains BD1 and BD2 (ovals), nuclear localization signal (NLS, triangle), and a C-terminal extraterminal (ET) domain (rectangle). (**B**) Schematic of the long (*brd2b-L*) and short (*brd2b-S*) transcript variants encoded by the *brd2b* locus, with numbered exons. UTRs of 5′ and 3′ are shown in white; protein-coding regions are shown in orange. The short transcript retains part of intron 5 (shown in yellow), which introduces a premature stop codon (darker yellow shows part of intron that is translated). Primer pairs flanking exon–exon junctions and used in RT-PCR are shown as dark blue arrows. (**C**) RT-PCR products produced from mRNA of staged oocytes (top gels) and 4, 24, and 48 hpf embryos (bottom gels), using various primer pairs, as shown in B. Lanes labeled with the same number display the PCR product obtained using the same primer pairs, as shown in the key: (1) 1 kb ladder (NEB); (2) ex5/in5 primers (*brd2b-S *250 bp predicted product); (3) ex5/6 primers (*brd2b-L* 400 bp predicted product); (4) ex9/10 primers (*brd2b-L* 300 bp predicted product); (5) ex11/12 (*brd2b-L *404 bp predicted product); (6) ex1/in5 primers (full-length *brd2b-S *1447 bp predicted product); (7) ex1/ex12 primers (full-length *brd2b-L *4037 bp predicted product); (8) primers for *β-actin* (+) control (322 bp predicted product); (9) primers internal to intron 1 (not shown in B), as (−) control for genomic DNA (784 bp predicted product). Maternal *brd2b-L* predicted PCR products are detected in all stages of ooctye tested (blue arrowheads, lanes labeled 3,4,5, top gels), but *brd2b-S* predicted products are not present (lanes labeled 2, top gels). The bands observed in stage 1, 2 and 3 oocytes in lanes labeled 2 are either smaller than the 250 bp predicted product for *brd2b-S* (s1, s2 gels), or are very faint products among several faint products (s3 gel), so are likely a spurious. Zygotic *brd2b-S* predicted product is detected in 24 hpf and 48 hpf embryos (red asterisk, lanes labeled 2, bottom gels), but not in 4 hpf embryos, while zygotic *brd2b-L* is detected in all stages of embryos tested (blue arrowheads, lanes labeled 3,4,5, bottom gels).

**Figure 2 jdb-09-00046-f002:**
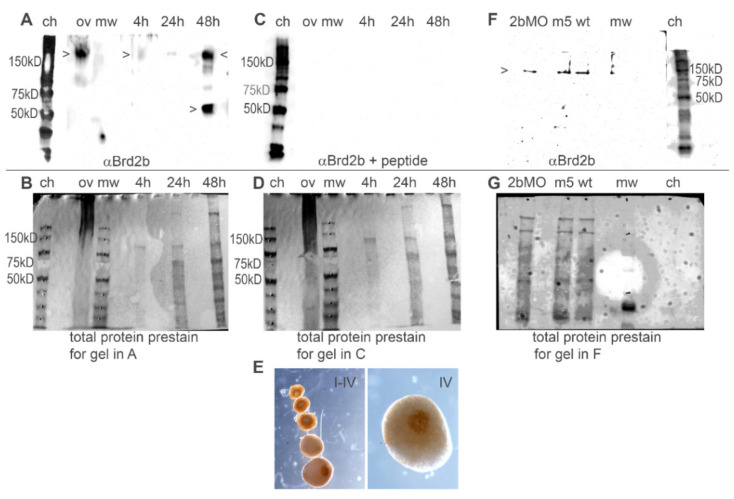
Anti-Brd2b peptide antibody detects major protein products in zebrafish embryos and oocytes, that are reduced in *brd2b*MO morphants. (**A**,**B**) Western blot (**A**) of total ovary (ov) and staged embryos (4 h, 24 h, 48 h) probed with anti-Brd2b peptide antibody. Blot was prestained (**B**) to assess total protein and relative loading across lanes. Total ovary samples often show a large amount of high molecular weight staining corresponding to yolk proteins that make up predominant products in this tissue; nevertheless, these do not interfere with the detection of Brd2b. A product over 150 kD is consistently detected in ovaries and 48 hpf embryos and sometimes faintly in 4 and 24 hpf embryos (top arrowheads in A); a product between 50 and 75 kD is also consistently detected in 48 hpf embryos (bottom arrowhead) and sometimes also in 4 hpf and 24 hpf embryos (see Appendix A). (**C**,**D**) Duplicate blot of A probed with anti-Brd2b peptide antibody in the presence of peptide antigen for peptide competition assay (**C**). The blot was prestained (**D**) to assess total protein and relative loading across lanes before antibody + peptide incubation. Peptide competition prevents all antibody signals on blot (**C**), supporting specificity of antibody for Brd2b epitope. Prestaining of blot using Memcode protein stain (**D**) assures that lack of anti-Brd2b signal in C is due to peptide competition rather than lack of protein in lanes. (**E**) Immunohistochemical analysis of staged oocytes (I–IV) and higher magnification stage IV oocyte (IV) using anti-Brd2b peptide antibodies, and showing eventual localization to the micropyle and future animal pole. (**F**,**G**) Western blot to assess efficacy of *brd2b*MO morpholino knockdown. Twenty-four hours post-fertilization embryo treatment groups: uninjected (wt), control 5-base mismatch morpholino-injected (mis5), and *brd2b*MO1-injected (2bMO), probed with anti-Brd2b peptide antibody (**F**). The blot was prestained (**G**) to assess total loading and relative loading across lanes. When relative loading is taken into account, there is a 3-fold reduction in Brd2b (~150 kD band) in *brd2b*MO-treated embryos (2ng total) as measured by ImageJ (FiJI) using band pixel densities (see Methods). Anti-Brd2B peptide antibody used in all immunoblot experiments in this study was raised against an antigenic epitope between NLS and ET (see Materials and Methods), and detects only the long isoform (Brd2b-L). Bottom edges of Western blots are indicated by horizontal line in (**A**, **C** and **F**). mw = Precision plus dual color molecular weight standard (BioRad); ch = Precision plus Western Chemiluminescent marker (BioRad).

**Figure 3 jdb-09-00046-f003:**
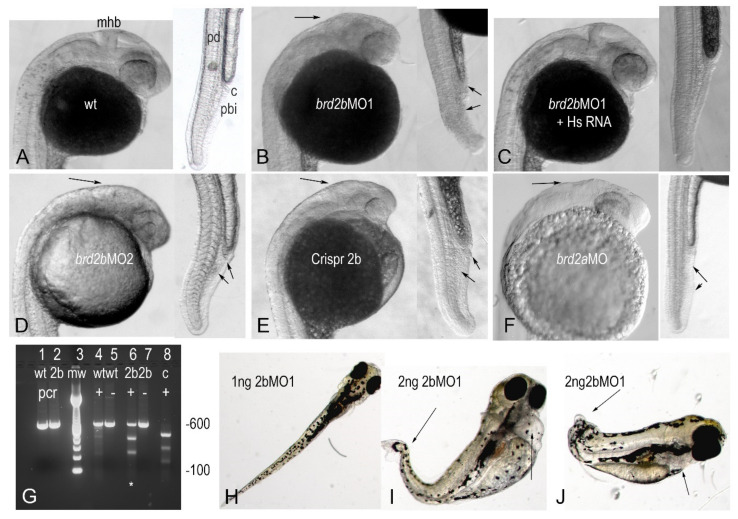
Brd2b knockdown results in reduced hindbrain, ill-defined MHB region, and trunk abnormalities similar to Brd2a morphants, but presents unique circulatory and pronephric defects. Brightfield images of 24 hpf prim 5 embryos (**A**–**F**), and larvae at 6 days (**H**–**J**). Lateral views of head and trunk of representative embryos from indicated treatment groups: (**A**) uninjected; (**B**) *brd2b*MO1-injected; (**C**) *Brd2*bMO1 + *HsBrd2*RNA co-injected; (**D**) *Brd2*MO2-injected; (**E**) Crispr-Cas9-*brd2b*-disrupted; and for comparison, (**F**) paralog *brd2a*MO-injected. In A) mhb = midbrain-hindbrain; pd = pronephric duct; c = cloaca; pbi = peripheral blood island. Head and trunk defects in morphants and Crispr-Cas9-treated embryos are indicated by black arrows, and include reduced brain and ill-defined MHB region, misformed pronephric duct, and disorganized PBI. These defects are substantially rescued by co-injection of human *Brd2* RNA, recapitulated by a non-overlapping anti-*brd2b* morpholino (MO2), and phenocopied by Crispr-Cas9 disruption, showing gene-specificity of observed effects. (**G**) Crispr-Cas9 *brd2b* target validation, showing intact PCR products from wildtype (lane 1, wt) and Crispr-disrupted (lane 2, 2b) *brd2b* locus; wildtype PCR product treated with mismatch-detecting resolvase (lane 4, wt+) and untreated (lane5, wt−); Crispr-disrupted PCR product treated with resolvase (lane 6, 2b+) and untreated (lane 7, 2b−). Crispr-disrupted *brd2b* locus treated with resolvase shows cleavage products (asterisk highlighting lane 6), while wildtype locus remains intact, indicating that Crispr-Cas9 targeted designated sequences in the *brd2b* locus. Pixel density of the 600 bp intact product band is reduced from 980 in untreated (lane 7) to 528 in treated (lane 6) Crispr-disrupted embryos, indicating close to 50% efficacy (see Methods). (**H**) Six day old morphant larvae injected at 2 cell stage with 1 ng *brd2b*MO1, or (**I**,**J**) 2ng *brd2b*MO1, illustrating dosage-dependent severe heart edema and trunk deformities, due to nearly complete lack of blood circulation. See Table 1 for population morphology data from these studies. See Appendix A for *HsBrd2* RNA control embryos images.

**Figure 4 jdb-09-00046-f004:**
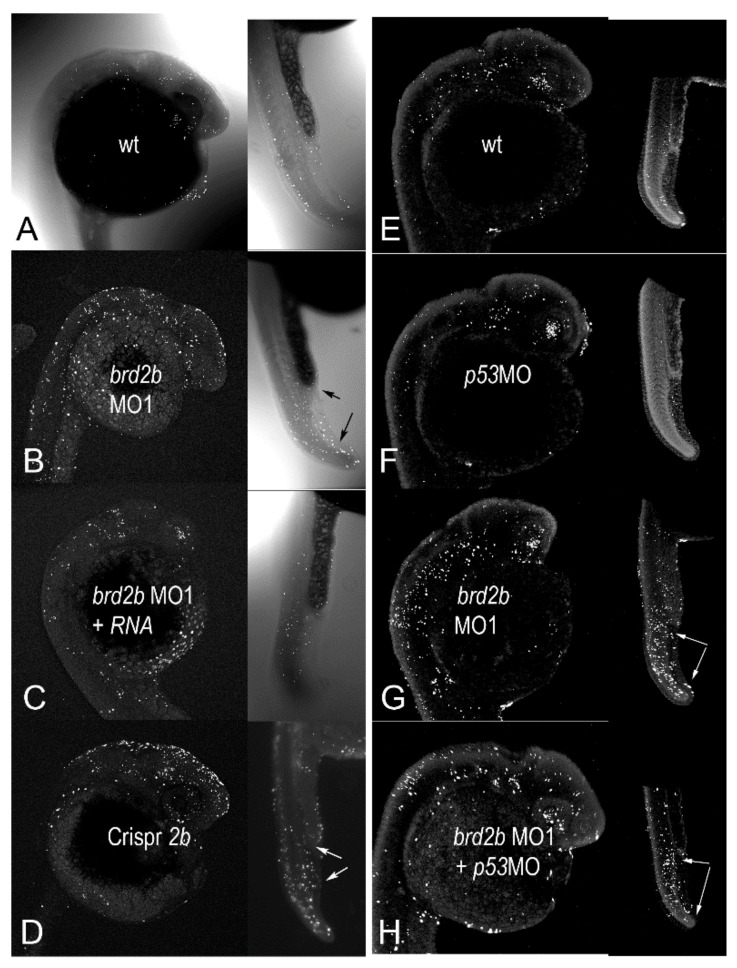
Brd2b knockdown increases cell death in the CNS of prim 5 morphant embryos but reduces cell death in the cloaca of the pronephros. Dark-field images of representative 24 hpf prim5 embryos treated as indicated and after TUNEL assay for apoptotic nuclei. Panels (**A**–**D**) (heads and trunks) show treatment groups for testing gene-specificity of apoptotic effects: (**A**) uninjected; (**B**) *brd2b*MO1-injected; (**C**) *brd2b*MO1 + *HsBrd2*RNA-injected; and (**D**) Crispr-Cas9-*brd2b*-disrupted embryos. *HsBrd2*RNA co-injection rescues, while Crispr-Cas9 treatment phenocopies, excess apoptosis in the brain and trunk overall of morphants, showing effects are specific to *brd2b*. Arrows indicate specific regions of *reduced* apoptosis at the cloaca (upper arrows) and *increased* apoptosis in the PBI (lower arrows) in the ventral trunk of morphant and Crispr-Cas9-treated, but not control or rescued embryos. Panels (**E**–**H**) (heads and trunks) show treatment groups for testing p53-dependent off-target effects: (**E**) uninjected; (**F**) *p53*MO-injected; (**G**) *brd2b*MO1-injected; and (**H**) *brd2b*MO1− + *p53*MO co-injected. *p53*MO co-injection does not abrogate excess apoptosis in morphants, ruling out off-target effects as the cause of cell death. See Figure 5 for quantitative TUNEL data from these studies. See Appendix A for images on *HsBrd2* RNA control embryos.

**Figure 5 jdb-09-00046-f005:**
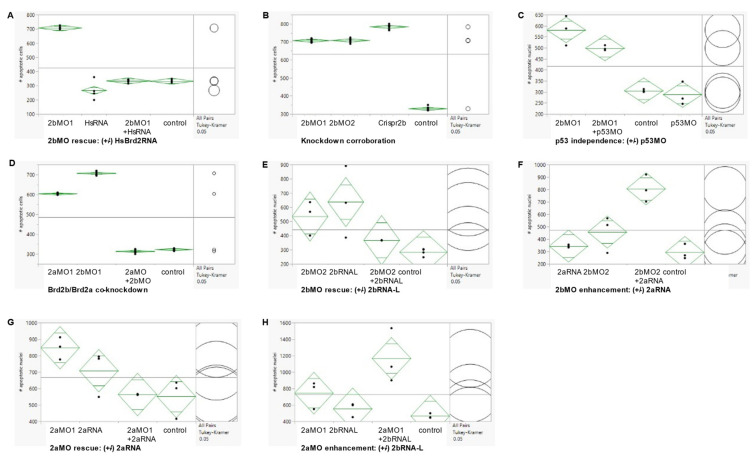
Quantitative TUNEL analysis of cell death: knockdown, co-knockdown, and antagonism studies. Cell death levels were measured in *brd2b*MO morphants and control embryos under various treatments using fluorescence TUNEL assay followed by quantitative laser-scanning confocal microscopy. The number of apoptotic nuclei in multiple optical sections from the brains of three to ten embryos per treatment, depending on experiment, were compared (see Methods for details). (**A**) RNA rescue treatments: *brd2b*MO1-injected (2bMO1), human *Brd2* RNA-injected (HsRNA), *brd2b*MO1 + human *Brd2* RNA-injected (2bMO+HsRNA), and uninjected (control). Means vary significantly by treatment (*p* < 0.0001, one-way ANOVA, Tukey’s HSD), with the greatest difference between 2bMO1 and all other treatments and no significant difference between “control” and “2bMO+HsRNA”, indicating effective rescue by exogenous human *Brd2* RNA. Cell death numbers for HsBrd2 RNA-injected controls were obtained from embryos from a separate clutch but reflect what we have consistently seen over multiple independent trials. (**B**) *brd2b*MO knockdown corroboration treatments: *brd2b*MO1-injected (2bMO1), *brd2b*MO2-injected (2bMO2), Crispr-Cas9-*brd2b* disruption (Crispr2b), and uninjected (control). Means vary significantly by treatment (*p* < 0.0001, one-way ANOVA, Tukey HSD), with Crispr2b, 2bMO1 and 2bMO2 all showing effects significantly greater than “control”; 2bMO1 and 2bMO2 show equivalent and moderate effects, while Crispr2b shows the greatest effect. Thus, three independent treatments targeting *brd2b* result in similarly increased cell death levels, showing gene-specificity of effect. (**C**) *p53*-dependent off-target effect treatments: *brd2b*MO1-injected (2bMO1), *brd2b*MO1 + *p53*MO co-injected (2bMO1 + p53MO), uninjected (control), and *p53*MO-injected (p53MO). Means vary significantly by treatment (*p* < 0.0001, one-way ANOVA, Tukey’s HSD), with no significant difference between 2bMO1 and “2bMO1 + p53MO” or between “control” and p53MO, but significant differences between the two pairs, indicating excess apoptosis is *brd2b*-specific, and not the result of p53-dependent off-target effects. (**D**) Co-knockdown treatments: *brd2a*MO1-injected (2aMO1), *brd2b*MO1-injected (2bMO1), *brd2a*MO1 + *brd2b*MO1 co-injected (2aMO+2bMO), and uninjected (control). Means vary significantly by treatment (*p* < 0.0001, one-way ANOVA, Tukey’s HSD), with “control” and “2aMO + 2bMO” not significantly different from each other and both significantly different from either 2bMO1 or 2aMO1; 2bMO1 shows a greater effect than 2aMO1. Simultaneous knockdown of both paralogs suppresses excess cell death observed in single knockdowns of either paralog and restores wild type levels of apoptosis. Panels (**E**–**H**) Rescue-Enhancement studies using zebrafish *brd2a* and *brd2b-L* in vitro-synthesized RNAs (not human *Brd2*) tested functional antagonism between paralogs at the level of apoptosis. (**E**): *brd2b*MO rescue treatments: *brd2b*MO2 (2bMO2), *brd2b*-*L*RNA (2bRNAL), *brd2bMO2* + *brd2b-L*RNA (2bMO2 + 2bRNAL), and uninjected (control). Means vary significantly by treatment (*p*-0.0286, one-way ANOVA, Tukey’s HSD), with “2bMO2+2bRNAL” closer to “control” in effect (*p* = 0.8407) than either 2bMO1 (*p* = 0.1319) or 2bRNAL (*p* = 0.0295), indicating partial rescue of the *brd2b*MO morphant defect. (**F**) *brd2b*MO enhancement treatments: *brd2a*RNA-injected (2aRNA), *brd2b*MO2-injected (2bMO2), *brd2a*RNA + *brd2b*MO2 co-injected(2bMO2+2aRNA), and uninjected (control). Means vary significantly by treatment (*p* = 0.0008, one-way ANOVA, Tukey’s HSD), with “2bMO2+2aRNA” giving significantly greater effect compared to “control” (*p* = 0.0009) than either 2bMO2 (*p* = 0.2405) or 2aRNA (*p* = 0.9151). (**G**) *brd2a*MO rescue treatments: *brd2a*MO1-injected (2aMO1), *brd2a*RNA-injected (2aRNA), *brd2a*MO1 + *brd2a*RNA co-injected (2aMO1 + 2aRNA), and uninjected (control). Means vary significantly be treatment (*p*
**=** −0.0176, one-way ANOVA, Tukey’s HSD), with “2aMO1 + 2aRNA” closer to “control” in effect (*p* = 0.9985) than either 2aRNA (*p* = 0.3286) or 2aMO1 (*p* = 0.0.0294), indicating partial rescue of the *brd2a*MO morphant defect. (**H**) *brd2a*MO enhancement treatments: *brd2a*MO1-injected (2aMO1), *brd2b-L*RNA-injected (2bRNAL), *brd2a*MO1 + *brd2b-L*RNA co-injected (2aMO1 + 2bRNAL), and uninjected (control). Means vary significantly by treatment (*p* = 0.0084, one-way ANOVA, Tukey’s HSD), with “2aMO1+2bRNAL” giving significantly greater effect compared to “control” (*p* = 0.0085), than either 2aMO (*p* = 0.3433) or 2bRNAL (0.9342). Note: for experiments in **A**, **B**, and **D**, apoptotic cell counts were obtained from 55 to 60 optical sections from 6 to 10 embryos per treatment; for experiment **C**, from maximum projection images from 3 embryos per treatment; for experiments (**E**–**H**), from 35 optical sections from 3 to 5 embryos per treatment. Green diamonds: confidence interval (95%) for group means with standard error. Black line: grand sample mean. Black circles: comparison circles for absolute differences of group means.

**Figure 6 jdb-09-00046-f006:**
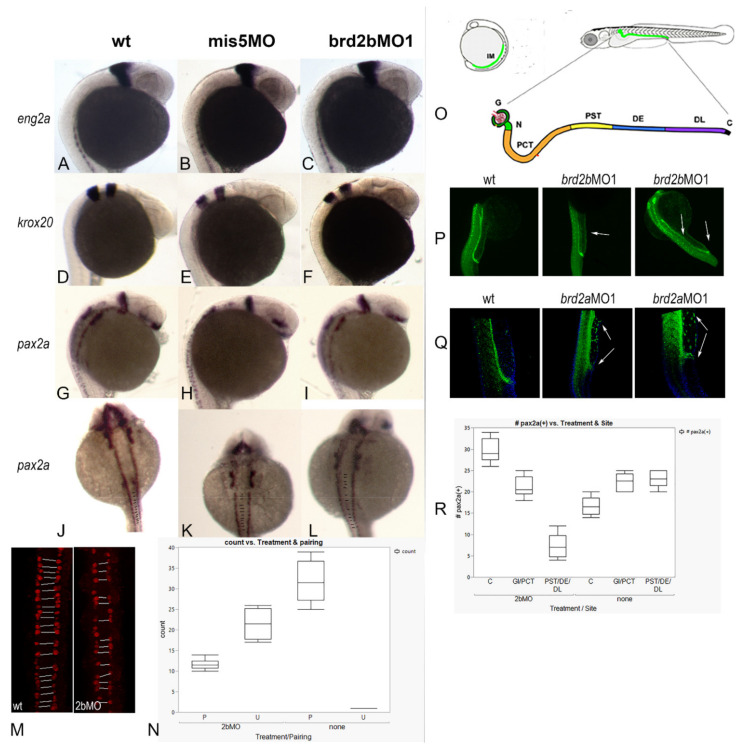
MHB expression of *pax2a, eng2a* and *krox20* mRNA is normal, but patterning of *pax2a* (+) spinal interneurons and pronephric cells is disrupted, in *brd2b* morphants. Panels (**A**–**L**): Brightfield images of representative 24 hpf prim5 control and morphant embryos assayed by In situ hybridization for expression of patterning genes at the MHB region (**A**–**I**) and in spinal interneurons (**J**–**L**). Uninjected embryos (wt; **A**,**D**,**G**,**J**), embryos injected with control *brd2b*-5-base mismatch morpholino (mis5MO; **B**,**E**,**H**,**K**), and embryos injected with *brd2b*MO (brd2bMO1; **C**,**F**,**I**,**L**) were assayed for *eng2a* (**A**,**B**,**C**), *krox20* (**D**,**E**,**F**,), and *pax2a* (**G**,**H**,**I**; **J**,**K**,**L**) expression. No differences in patterning gene expression at the MHB region are detected. Using *pax2a*(+) as a marker for spinal interneurons reveals mispairing on either side of the midline in morphants (**J**,**K** vs. **L**, gaps between lines). (**M**) Immunofluorescence with anti-Pax2a antibodies to visualize interneurons on either side of spinal cord, dorsal views of maximum projection confocal images. Representative control uninjected (wt) and *brd2b*MO1-injected (2bMO) embryos are shown with symmetrically paired interneurons linked by white lines; unpaired interneurons appear solo on one side of the central spinal cord. (**N**) The number of paired “P” vs. unpaired “U” interneurons in six embryos per treatment group (uninjected controls = none; *brd2b*MO-injected = 2bMO) was analyzed by chi-square contingency and is shown as a box plot (*p* < 0.0001). While, on average, only 3% of interneurons are unpaired in wild type, about 65% are unpaired in *brd2b*MO morphants. Panels (**O**–**R**): Analysis of Pax2a(+) pronephric precursor cells in the developing duct in *brd2b*MO and *brd2a*MO morphants by quantitative immunofluorescence. (**O**) Diagram of pronephric duct segments: G, glomerulus; N, neck; PCT, proximal convoluted tube; PST, proximal straight tube; DE, distal early tube; DL, distal late tube; C, cloaca. (**P**) Pax2a(+) immunofluorescence in uninjected (wt) and *brd2b*MO morphants (*brd2b*MO1). (**Q**) Pax2a(+) immunofluorescence in uninjected (wt) and *brd2a* morphants (*brd2a*MO1). Paralogs show opposite effects on distribution of Pax2a(+) pronephric cells (arrows). *brd2b* morphants show fewer cells in the more distal parts of the tube (PST, DE, DL) but more cells at the very end cloaca (C), compared to wild type, while *brd2a* morphants show excess cells along the tube, and a normal number of cells at the cloaca. (**R**) Quantitation of distribution of Pax2a(+) cells in uninjected (none) and *brd2b*MO1-injected embryos (2bMO), by pronephric segment. Boxplot of ANOVA and paired *t*-tests shows significant differences in the predicted direction between these two treatment groups in segments PST/DE/DL (*p* < 0.0001,) and in C (*p* < 0.0001), but not in G/PCT (*p* = 0.3971).

**Figure 7 jdb-09-00046-f007:**
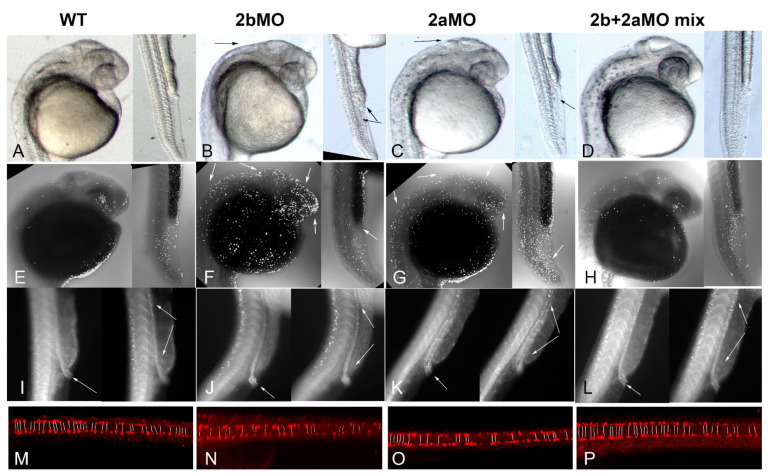
Co-knockdown of Brd2a and Brd2b suppresses morphant defects in both brain and pronephros of morphant embryos at 24 hpf. Double knockdown was used to test for genetic interaction between paralogs. Panels (**A**–**D**): Brightfield images of head and trunk of representative 24 hpf prim5 embryos from indicated treatment groups assessed for morphology: (**A**) uninjected, (**B**) *brd2b*MO-injected, (**C**) *brd2a*MO-injected, and (**D**) *brd2b*MO + *brd2a*MO co-injected. Panels (**E**–**H**): Darkfield images of representative embryos from the same treatment groups assessed by TUNEL assay for apoptotic nuclei: (**E**) uninjected, (**F**) *brd2b*MO-injected, (**G**) *brd2a*MO-injected, and (**H**) *brd2b*MO + *brd2a*MO co-injected. Co-injection of MOs from each paralog suppresses morphant defects in both morphology and cell death in the brain and trunk, indicating genetic interaction with functional antagonism between paralogs. Black arrows indicate reduced brain, ill-formed MHB region, disorganized PBI common to both morphants (**B**,**C**), and plug of excess cells at the cloaca unique to *brd2b*MO morphants (upper arrow in B, trunk). White arrows show increased apoptosis in brain and PBI common to both morphants (**F**,**G**, head; **G**, trunk), while lack of apoptosis is seen at the cloaca only in *brd2b*MO morphants (**F**, trunk). See Table 1 for population morphology data and Figure 5 for quantitative TUNEL data. Panels (**I**–**L**): Darkfield images of pronephric duct in trunks of representative 24 hpf prim 5 embryos from the indicated treatment groups assessed for Pax2a(+) cells by immunofluorescence**:** (**I**) uninjected, (**J**) *brd2b*MO-injected, (**K**) *brd2a*MO-injected, and (**L**) *brd2b*MO + *brd2a*MO co-injected. Cloacal opening is indicated by white arrow in left panels. Distribution along the ductal tube of Pax2a(+) pronephric precursor cells is highlighted between white arrows in right panels. Co-knockdown suppresses morphant defects of each paralog, bringing distribution of cells along the duct and at the cloaca closer to wildtype. (see also Figure 6O,P,Q). Panels (**M**–**P**) Darkfield images of spinal interneurons along dorsal trunks of representative 24 hpf embryos from the indicated treatments groups assessed for Pax2a(+) cells by immunofluorescence: (**M**) uninjected, (**N**) *brd2b*MO-injected, (**O**) *brd2a*MO-injected, and (**P**) *brd2b*MO + *brd2a*MO co-injected. Co-knockdown suppresses the morphant mis-pairing defect of each paralog, and restores interneuron number and pairing to near wildtype levels. Lines connect symmetrically paired neurons; unpaired neurons appear solo on one or the other side of the central nerve cord.

**Figure 8 jdb-09-00046-f008:**
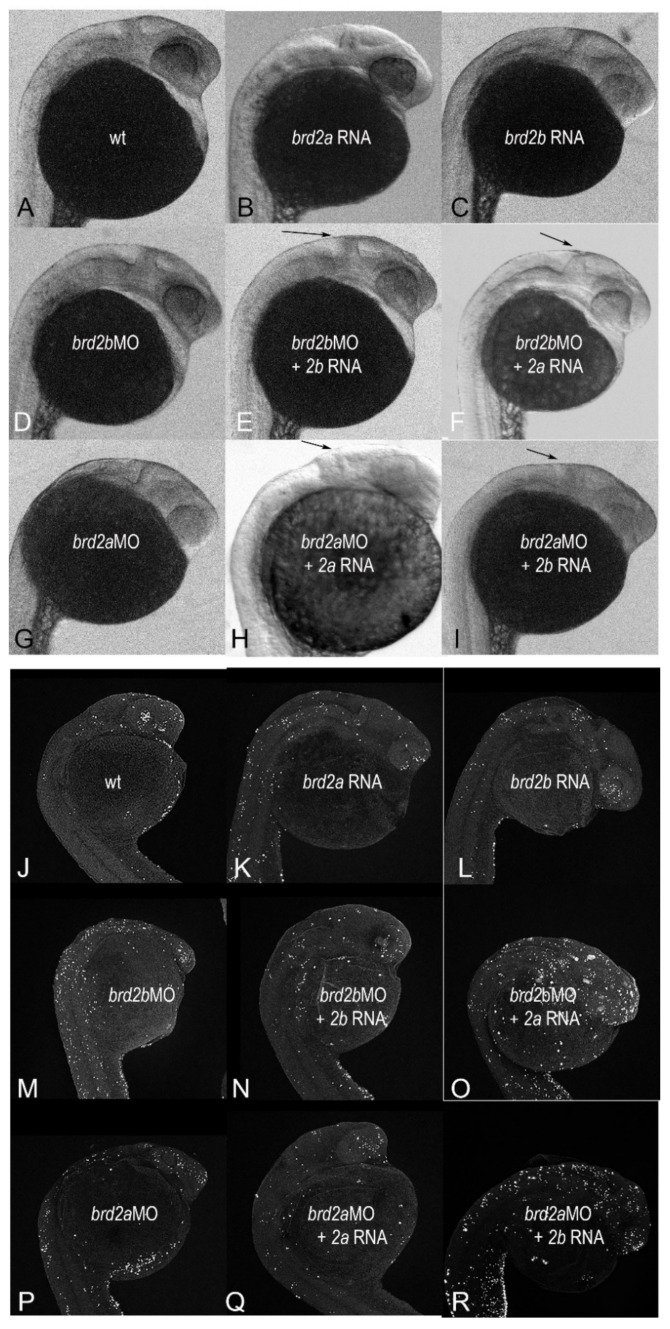
Enhancement of morphant brain phenotypes by injection of paralogous RNA corroborates genetic antagonism between *brd2a* and *brd2b.* Rescue-Enhancement studies using zebrafish *brd2a* and *brd2b-L* in vitro-synthesized RNAs (rather than human *Brd2*) were used to test functional antagonism between paralogs implied by double knockdown suppression. Panels (**A**–**I**): Brightfield images of representative 24 hpf prim 5 embryos from indicated treatment groups. For controls: (**A**) uninjected wild type, (**B**) *brd2a*RNA-injected, and (**C**) *brd2b-L*RNA-injected. For *brd2b*MO rescue-enhancement: (**D**) *brd2b*MO-injected, (**E**) *brd 2b*MO + *brd2b-L*RNA co-injected (rescue), and (**F**) *brd2b*MO + *brd2a*RNA co-injection (enhancement). For *brd2a*MO rescue-enhancement: (**G**) *brd2a*MO-injected, (**H**) *brd2a*MO + *brd2a*RNA co-injected (rescue), and (**I**) *brd2a*MO + *brd2b-L*RNA co-injection (enhancement). Arrows indicate rescue (**E**,**H**) and enhancement (**F**,**I**) of morphant brain defects by co-injection of cognate or paralogous RNA, respectively. See Table 2 for population morphology data. Panels (**J**–**R**): Darkfield images of representative 24 hpf prim 5 embryos from the same treatment groups after TUNEL assay: (**J**) uninjected wildtype, (**K**) *brd2a*RNA-injected, and (**L**) *brd2b-L*RNA-injected; (**M**) *brd2b*MO-injected, (**N**) *brd2b*MO + *brd2b-L*RNA co-injected (rescue), and (**O**) *brd2b*MO + *brd2a*RNA co-injection (enhancement); (**P**) *brd2a*MO-injected, (**Q**) *brd2a*MO + *brd2a*RNA co-injected (rescue), and (**R**) *brd2a*MO + *brd2b*RNA co-injection (enhancement). Note rescue (**N**,**Q**) and enhancement (**O**,**R**) of morphant excess apoptosis by co-injection of cognate or paralogous RNA, respectively, supporting the idea of functional antagonism between paralogs. See Figure 5 for quantitative TUNEL data.

**Table 1 jdb-09-00046-t001:** Population morphology data: RNA rescue, knockdown and co-knockdown studies.

Treatment	% Brain Defect ^a^	% Pbi Defect ^b^	% Duct Defect ^c^	% Circ Defects ^d^
*brd2bMO1 +/− HsBrd2RNA rescue ^e^*				
control	12.5	0	0	0
HsBrd2RNA	11.5	0	7.7	0
brd2bMO1	100	100	100	50
brd2bMO1 + HsBrd2RNA	20	8.6	17.1	5.7
Chi-square Contingency, Fisher’s Exact Test: *p* < 0.0001 for all four defects. Correspondence Analysis: [wildtype: control, RNA, brd2bMO+HsRNA] [defect: 2bMO], for all four parameters.				
*brd2b*MO *single knockdown corroboration ^f^*				
control	0	0	0	0
mis5	0	0	0	0
2bMO1	100	25.8	88.7	93.7
2bMO2	100	24.2	100	100
Crispr2b	100	21	100	100
Chi-square Contingency, Fisher’s Exact Test: *p* < 0.0001 for all four defects. Correspondence Analysis: [wildtype: control, mis5] [defect: MOs, Crispr] for brain; [wildtype: control, mis5] [defect: MOs, Crispr] for PBI, duct, circulation				
*brd2a*MO/*brd2b*MO *co-knockdown ^f^*				
control	0	0	0	0
2aMO1	100	80	0	0
2bMO1	100	26.7	91.7	96.7
2aMO1 + 2bMO1	5	0	0	0
Chi-square Contingency, Fisher’s Exact Test: *p* < 0.0001 for all four defects Correspondence Analysis: [Wildtype: control, 2aMO+2bMO] [Defect: 2aMO, 2bMO] for brain;				
[Wildtype: control, 2aMO+2bMO, 2bMO] [defect: 2aMO] For PBI; [Wildtype: control, 2aMO+2bMO, 2aMO] [defect: 2bMO] for duct, circulation				

^a^ reduced brain, ill-defined mhb region ^b^ disorganized or vacuous pbi ^c^ plugged or enlarged cloaca ^d^ severely reduced or absent circulation ^e^ 20–40 embryos per treatment; 2bmo, 2 ng; 2amo, 4 ng; hsbrd2 rna 250 pg ^f^ 60 embryos per treatment; 2bmo, 2 ng; 2amo, 4 ng.

**Table 2 jdb-09-00046-t002:** Population morphology data: Brd2a-Brd2b rescue-enhancement studies.

Rescue Treatment ^a^	% Defects ^b^
*2bMO rescue-2bRNA-L_**brain** ^c^*	
control	13.0
2bMO2	**40.4**
2bRNA-L	39.7
brd2bMO2 + 2bRNA-L	**24.6**
Chi-square Contingency, Fisher’s Exact Test: *p* = 0.0026 Correspondence Analysis: [wildtype: control, 2bMO2+2bRNA-L] [defect: 2bMO2, 2bRNA-L]	
*2aMO rescue-2aRNA_**brain** ^d^*	
Control	5
2aMO1	**35**
2aRNA	20
2aMO1 + 2aRNA	**20**
Chi-square Contingency, Fisher’s Exact Test: *p* = 0.1416 Correspondence Analysis: [wildtype: control, 2aMO1+2aRNA, 2aRNA] [defect: 2aMO1]	
*2bMO rescue-2bRNA-S_**brain** ^e^*	
control	5
2bMO2	**33.3**
2bRNA-S	25
2bMO2 + 2bRNA-S	**33**
Chi-square Contingency, Fisher’s Exact Test: *p* = 0.0840 Correspondence Analysis: [wildtype: control] [defect: 2bRNA-S, 2bMO2+2bRNA-S, 2bMO2]	
*2bMO rescue-2bRNA-L_**duct** ^f^*	
control	40
2bMO2	**80**
2bRNA-S	55
2bMO2 + 2bRNA-S	**29.6**
Chi-square contingency, Fisher’s Exact Test: *p* = 0.0128 Correspondence Analysis: [wildtype: control, 2bMO2+2bRNA-S] [defect: 2bRNA-S, 2bMO]	
**Enhancement Treatment ^g^**	**% Defects ^b^**
*2bMO enhancement-2aRNA_**brain**^f^*	
control	11.1
2bMO2	**30.8**
2aRNA	25
2bMO2 + 2aRNA	**42.9**
Chi-square contingency, Fisher’s Exact Test: *p* = 0.0059 Correspondence Analysis: [wildtype: control, 2aRNA][2bMO][defect: 2bMO2+2aRNA]	
*2aMO enhancement- 2bRNA-L_**brain** ^f^*	
control	11.5
2aMO1	**41.5**
2bRNA-L	30.4
2aMO1 + 2bRNA-L	**61.4**
Chi-square contingency, Fisher’s Exact Test: *p* = 0.0109 Correspondence Analysis: [wildtype: control, 2bRNA-L][2aMO][defect: 2aMO1+2bRNA-L]	
*2aMO enhancement 2bRNA-S_**brain**^e^* control 2aMO1 2bRNA-S 2aMO1 + 2bRNA-S	
	0
	**63.6**
	50
	**71.4**
Chi-square contingency, Fisher’s exact test: *p* < 0.0001 Correspondence analysis: [wildtype: control][2bRNA-S][defect: 2aMO, 2aMO1+2bRNA-L]	
*2bMO enhancement-2aRNA_**duct**^c^* Control 2bMO2 2aRNA 2bMO2 + 2aRNA	
	16.2
	**48.2**
	50
	**69.2**
>Chi square contingency, Fisher’s exact test: *p* < 0.0001 Correspondence analysis: [wildtype: control][2aRNA, 2bMO] [defect: 2bMO2+2aRNA]	

^a^ 2bMO, 4 ng; 2aMO, 4 ng; RNA, 200 pg; grey heading indicates no rescue ^b^ Only moderate severity defects were counted for these analyses, to avoid ambiguities that may arise in scoring between wildtype and mild defects. Bold numbers call attention to pairs directly involved in rescue or enhancement comparisons. ^c^ 50–60 embryos per treatment ^d^ 20 embryos per treatment ^e^ 15–30 embryos per treatment ^f^ 35–50 embryos per treatment ^g^ 2bMO, ~3 ng; 2aMO, ~3ng; RNA treatments, 200 pg.

**Table 3 jdb-09-00046-t003:** Summary of paralog interaction with functional antagonism.

Morphant Defects	Brd2a Deficiency	Brd2b Deficiency	Suppression	Skew from WT	Path ^a^
	2AMO	2BRNA-L	2BRNA-S	2BMO	2ARNA	2AMO + 2BMO	AA:AB:BB	AA:AB:BB	
								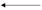	
**CNS**									
Brain Morphology/PCD	X	X		X	X	WT	AA < AB < BB or AA > AB > BB	pcd+
MHB Genes	X			WT		WT	AA < AB < BB		hox
VNC Interneurons	X			X		WT	AA < AB < BB or AA > AB > BB	hox
**PBI**									
Morphology/PCD	X			X		WT	AA < AB < BB or AA > AB > BB	pcd+
**Pronephric Duct**									
Tube PAX2a(+)	X (+)			X (−)			AA < AB < BB		lin+ m
								AA > AB > BB	lin− m
**Cloaca**									
Plug	WT		X ^b^	X	X	WT		AA > AB > BB ^b^	pcd+ m
PCD	WT			X (−)		WT		AA > AB > BB	pcd−
PAX2a (+)	WT			X (+)		WT		AA > AB > BB	pcd− m
**Circ/Lethality**				X		WT		AA > AB > BB	???

^a^ potential pathway involved: pcd = cell death; hox = homeobox; lin=lineage specification; m = mitosis or migration. ^b^ red X indicates *brd2b*-*s* rna short variant; red “B” indicates Brd2b-S protein short isoform.

## Data Availability

The data presented in this study are available in the article itself and the Appendix A provided.

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
