# Peer review of "Zebrafish Paralogs brd2a and brd2b Are Needed for Proper Circulatory, Excretory and Central Nervous System Formation and Act as Genetic Antagonists during Development"

_jdb, 2021, doi:10.3390/jdb9040046_

Round 1

Reviewer 1 Report

It is important to understand how gene duplication and divergence might contribute to the evolution of gene networks and developmental pathways. Zebrafish is an excellent model system to study the genetic interaction between paralogs in development. The paper titled “Zebrafish paralogs brd2a and brd2b are needed for proper circulatory, excretory and central nervous system formation and act as genetic antagonists during development” shows that BET family paralogs brd2b and brd2a interact genetically as functional antagonists during zebrafish development to effect proper formation of the brain, in part through their co-regulation of cell death. Even though this paper shows interesting data, many aspects of this paper should be improved to make this paper more comprehensive and informative that could be appreciated by the research community. Overall, the text requires major revisions, and the figures need to be re-organized to maintain the scientific flow.

Here are my major comments to authors.

  1. Figure 1C should be replaced with high resolution gel images and the figure label on the right should be replaced with clear letters. The submitted original figures look good in resolution. Corresponding lanes of gel image from 24 hpf has not been labeled, consistent with other gels. Figure legend says “Zygotic brd2b-S product is detected in 24 hpf and 48 hpf embryos (red circles, lane 2)” in line 445 and 446. But in 24 hpf, lane 3 from the ladder is circled in red.
  2. Figure 2E is a repeat of Figure 2A, and why this is included in the main figure? The blots are pre-stained to assess relative loading across lanes, the text and the figure legend corresponding these figures are not clearly explained. Can the authors elaborate more on this about what the reader should look for? Some blots are presented with the mw marker, some are not. The brightness of some blots is at the extreme end compared to others, make it difficult to compare.
  3. In figure 2, the authors state that the post-translational modification or stable complex formation could be the reason for observed difference for Brd2b-L in Western blots compared to the predicted size. Are BET family proteins known to be post-translationally modified or in complexes with other proteins in other organisms?  
  4. In Figure 3, the knock down efficiencies of 2bMO2, Crispr2b and brd2aMO have neither been shown nor discussed/compared with 2bMO1 shown in figure 2.
  5. Can the authors provide a quantitative representation of apoptotic nuclei from uninjected, brd2bMO1-injected, brd2bMO1 + HsBrd2RNA-injected and Crispr-Cas9-brd2b-disrupted embryos in Figure 4 in total (the authors provided quantitative data for apoptotic nuclei localized to post-anal tail, PBI, and dorsal spinal cord)? The dark-field images suggest that HsBrd2RNA injected into brd2bMO1 did not rescue the apoptotic phenotype to the uninjected (wild type) levels in Figure 4, however it rescued to the uninjected levels in 5A. Can the authors comment on this?
  6. Figure 7E-7H is a repetition of Figure 4D. The authors should re-organize all the figures to avoid repetition and to maintain the scientific flow of the data.
  7. Can the authors speculate any reasons for why they observed that the injection of either zebrafish brd2b-LRNA or brd2aRNA alone into embryos often results in mild brain defects that phenocopy morphant defects in Figure 8?
  8. The discussion section is loaded with information that can/should be included in the introduction section. Discussion needs to be re-written to summarize the overall discoveries of the paper and should discuss the questions that the reader might have after reading the results section.

Minor comments:

  1. Typo in line 110, ‘roll’ instead of ‘role’
  2. Inconsistency in writing temperatures in the methods section. For example, 4°C and 4C are used.
  3. Gap in line 198.
  4. Gap between “ApopTag” and “Fluorescein” in line 363.

Reviewer 2 Report

The authors have studied the Brd2 paralogs Brd2a and Brd2b in zebrafish development. The study is well conducted and nicely written. I like how the authors discuss various possibilities and even compare the similarities between the Brd isoforms in other species (like fly for example). The article includes a lot of useful information and is useful to scientific community. I have some minor suggestions. 

  1. In Figure 1C, authors claim that Brd2b-S is only present in late stage embryos (lane 2; 24hpf and 48hpf). I think the band is also lightly visible in S2 and S3 oocytes? Is that just background or can it be lower yield on PCR? Please clarify. Better quality gel images and maybe replacing red boxes with white arrows in the image will help readers interpret data better. 
  2. The molecular weight of protein ladders needs to be in all western blots in figure 2. The quality of the gels are bad and needs to be improved to show clear bands.
  3. Separate loading control of western blots needs to be used. Also authors can indicate what primary antibody they used for each blot next to the blot. It will be easier than having to read figure legend every time. 

Round 2

Reviewer 1 Report

The authors have addressed all the major and minor comments with appropriate figures (replaced) and text. The confusing parts from the original submission have re-written to provide more clarity to the readers. Even though the authors refused to address one of my major comments, they have clearly justified why they didn't want to make a change. I think the manuscript is more comprehensive and with higher scientific soundness compared to the previous version. 

Author Response

We would like to thank this reviewer for useful comments and acceptance of our arguments for leaving the discussion as written. We appreciate the time and effort the reviewer gave our paper.